



# Developing a Parsimonious Canopy Model (PCM v1.0) to Predict Forest Gross Primary Productivity and Leaf Area Index

Bahar Bahrami [*1], Anke Hildebrandt[1,2], Stephan Thober[1], Corinna Rebmann[1], Rico Fischer[3], Luis Samaniego[1], Oldrich Rakovec[1,4], and Rohini Kumar[1]

[1] Department of Computational Hydro-system, Helmholtz Centre for Environmental Research-UFZ, Leipzig, Germany

[2] Friedrich Schiller University Jena, Institute of Geoscience, Terrestrial Ecohydrology, Burgweg 11, 07745 Jena, Germany

[3] Department of Ecological Modelling, Helmholtz Centre for Environmental Research-UFZ, Leipzig, Germany

[4] Faculty of Environmental Sciences, Czech University of Life Sciences Prague, Praha-Suchdol 16500, Czech Republic

**Correspondence:** Bahar Bahrami (bahareh.bahrami@ufz.de)

**Abstract.** Temperate forest ecosystems play a crucial role in governing global carbon and water cycles. However, unprecedented global warming poses fundamental alterations to forest ecological functions (e.g. carbon uptake) and forest biophysical variables (e.g. leaf area index). Quantification of forest carbon uptake, gross primary productivity (GPP), as the largest carbon flux has a direct consequence on carbon budget estimations. Part of this assimilated carbon stored in leaf biomass is related to

the leaf area index (LAI), which is of critical significance in and closely linked to water cycle. There already exist a number of models to simulate dynamics of LAI and GPP, however, the level of complexity, demanding data, and poorly known parameters often prohibit the model applicability over data-sparse and large domains. In addition, the complex mechanism associated with coupling the terrestrial carbon and water cycles poses a major challenge for integrated assessments of interlinked processes (e.g. accounting for temporal dynamic of LAI for improving water balance estimations and soil moisture availability

for enhancing carbon balance estimations). In this study, we propose a parsimonious forest canopy model (PCM) to predict daily dynamics of LAI and GPP with few required input which is also suitable for integration into state-of-the-art hydrologic models. The light use efficiency (LUE) concept is central to PCM (v1.0), coupled with a phenology submodel. PCM estimates total assimilated carbon based on conversion efficiency of absorbed photosynthetically active radiation into biomass. Equipped with the coupled phenology submodel, the total assimilated carbon partly converts to leaf biomass from which prognostic and

temperature-driven LAI is simulated. The model combines modules for estimation of soil hydraulic parameters based on the so-called pedotransfer functions and vertically weighted soil moisture considering the underground root distribution, when soil moisture data is available. We test the model on deciduous broad-leaved forest sites in Europe and North America selected from the FLUXNET network. We analyze the model parameter sensitivity on the resulting GPP and LAI and identified on average 10 common sensitive parameters at each study site (e.g., LUE, SLA, etc). Model performance is evaluated in a verification

period using in situ measurements of GPP and LAI (when available) at eddy covariance flux towers. The model adequately captures the daily dynamics of observed GPP and LAI at each study site (Kling-Gupta-Efficiency; KGE varies between 0.79 and 0.92). Finally, we investigate the cross-location transferability of model parameters and derive a compromise parameter set to be used across different sites. The model also showed robustness with the compromise single set of parameters, applicable

---

*Corresponding author (bahareh.bahrami@ufz.de)





to different sites, with an acceptable loss in model skill (on average $\pm$ 8%). Overall, in addition to the satisfactory performance
of the PCM as a stand-alone canopy model, the parsimonious and modular structure of the developed PCM allows for a smooth
incorporation of carbon modules to existing hydrologic models. Thereby, it facilitates the seamless representation of coupled
water and carbon cycle components, i.e. prognostic simulated vegetation leaf area index (LAI) would improve the representa-
tion of the water cycle components (e.g., evapotranspiration), while GPP predictions would benefit from simulated soil water
storage from a hydrologic model.

# 1  Introduction

As the climate is changing, the future functionality and resilience of terrestrial ecosystems are expected to change in numerous
ways. Fundamentally, terrestrial ecosystems drive the life-sustaining exchanges of matter and energy between land and at-
mosphere (e.g., carbon dioxide/water vapor exchange). However, increased concentrations of greenhouse gases and projected
global warming (IPCC, 2021), contribute to unprecedented extreme climate events and changes in ecosystem functioning and
productivity (Malhi et al., 2020). Depending on the frequency and intensity of extreme events together with other aspects
of anthropogenic change, ecosystem patterns and processes such as carbon dioxide uptake and water vapour release can be
altered, potentially irreversibly (Grimm et al., 2013). Given the importance of carbon dioxide as a principal greenhouse gas
that drives global climate change and the extent to which ecosystems are capable of sequestrating it, there has been growing
attention toward the quantification of carbon fluxes/stocks and understanding the role of the terrestrial ecosystems in regulating
the exchange of carbon between land and atmosphere (Beer et al., 2010). Temperate forest ecosystems, including deciduous
broad-leaved forest (DBF), are known as an integral part of global carbon cycle and contribute to climate change mitigation
by removing carbon from the atmosphere (Reinmann and Hutyra, 2017). Forests are recognized as biomes with high carbon
sequestration capacity (Lal and Lorenz, 2012) equivalent to around 60% of the global net forest sink (Reinmann and Hutyra,
2017). The total carbon uptake from the atmosphere into the ecosystems by plant photosynthesis is known as vegetation gross
primary production (GPP). GPP is the largest flux within the carbon cycle (Schaefer et al., 2012; Foley and Ramankutty, 2003)
and has a direct effect on moderating climate and environment by sequestering anthropogenically emitted $CO_2$. In turn, these
ecosystems are also vulnerable to the adverse effects of changing climate (Luyssaert et al., 2007; Senf et al., 2018; Forzieri
et al., 2021) where the spatial pattern of vulnerability is controlled by environmental conditions (Forzieri et al., 2021), Specif-
ically, environmental constraints, such as temperature, water availability, and radiation control vegetation productivity and
regulate the rate of GPP (Yuan et al., 2014). The DBF biomes are characterized by favourable climate in four distinct seasons
with a temperature-driven canopy structure. The plant canopy capacity and seasonality are expressed by leaf area index (LAI)
(Wang et al., 2019). The LAI is defined as one half the total green leaf area per unit horizontal ground surface area. LAI can
be estimated by field measurements and be simulated by the fractional accumulated carbon stored in the leaf pool within the
carbon cycle. This key biophysical plant variable affects not only sequestering carbon from the atmosphere via photosynthesis
but also the release of water to the atmosphere through transpiration (Yang et al., 2017). However, accounting for a dynamic
representation of vegetation characteristics (e.g., leaf area index) relevant for accurate estimation of water balance components





(e.g., plant transpiration) in most of conceptual hydrologic models is not properly considered, especially for the assessment of climate change impacts on water balance components (Wegehenkel, 2009; Asaadi et al., 2018). Many models have been successfully developed to estimate GPP, spanning a range of complexity and representation of physical and biological processes

(Che et al., 2014; Arora, 2002; Ostle et al., 2009). The GPP models are generally divided into three categories of empirical, enzyme kinetic (EK), and light use efficiency (LUE) models (Schaefer et al., 2012). The first category, the empirical models, are data-oriented approaches where statistical relationships between inferred GPP from flux observations (eddy covariance - EC) and observed environmental conditions are established; and those inferred relationships are then expanded to large scales ranging from regional to global levels (Beer et al., 2010; Schaefer et al., 2012). The second category, the enzyme kinetic (EK)

approach, represents leaf scale GPP as a result of a complex set of biophysical and biochemical reactions. This includes first, the light reaction (in which light energy splits water molecules, travelling from the soil to leaf chloroplasts, into $O_2$, electrons, and $H^+$ to produce electron carrier molecule (the reduced form of nicotine adenine dinucleotide phosphoric acid; NADPH) and energy storage (adenosine triphosphate; ATP). Later, in the next step for dark reaction (Calvin cycle), the rubisco enzyme uses the ATP energy from the light response to sequestrate the atmospheric $CO_2$ into organic carbon (Farquhar et al., 1980; Collatz

et al., 1992). This approach requires the specification of a relatively large number of parameters for governing processes, whose acquisition (through direct measurements) is not straightforward - specifically at scales larger than the leaf level. Finally, the last category for the GPP estimation is a widely used approach based on the light use efficiency (LUE) concept, relevant for its applications over larger scales at regional and global (Potter et al., 1993; Yuan et al., 2007). By implementing simplified relationships that hold at the ecosystem level and avoiding a detailed parameterization of leaf-level processes, the LUE concept

is particularly relevant for quantifying carbon budget at landscape and larger scales (Street et al., 2007; Wei et al., 2017). In this approach, ecosystem GPP is related to absorbed photosynthetically active radiation (APAR) as a product of incident photosynthetically active radiation (PAR) by a fraction of PAR (fPAR) absorbed by plant leaves through a biome specific LUE parameter. The LUE is a vegetation conversion efficiency factor of absorbed radiation into biomass and is defined as the amount of carbon produced per unit of absorbed PAR (Yuan et al., 2014). The amount of sequestrated carbon as biomass will then be allocated

to different plant carbon pools (i.e. leaf, stems, and roots) controlled by the relative demand exerted by these pools at different periods (Arora, 2002). Several LUE models have been successfully applied for estimating the ecosystem GPP such as, CFLUX (Turner et al., 2006), EC-LUE (Yuan et al., 2007), MODIS-GPP (Running et al., 2004), VPM (Xiao et al., 2004), and CASA (Potter et al., 1993) at different spatial and temporal scales (Wei et al., 2017; Law et al., 2000; Coops et al., 2005). However, despite the large potential of these LUE models, they are highly dependent on satellite-based data such as remotely sensed LAI

and fPAR (Wang et al., 2017). These two key biophysical variables are generally sensitive to cloud contamination (Chen et al., 2019; Zhu et al., 2013) leading to temporal and spatial gaps, particularly over areas like central Europe. It has also been documented that the satellite-based fPAR and LAI products are subjected to uncertainty, and that may induce errors in quantifying GPP (Wang et al., 2017). Overall, several factors, including either high demand of required data and computation in detailed models or dependency of simplified LUE models on satellite data might hinder the application of existing models. Concerning

estimation of LAI and its impact on water balance, the utilization of carbon uptake and leaf biomass growth has been used in the TETIS-VEG ecohydrology model (Francés et al., 2007; Pasquato et al., 2015). The TETIS-VEG model is however adapted





for the evergreen forest, thus it is not applicable over deciduous broad-leaved forest with a distinct seasonal leaf development dynamic. Another approach to simulate GPP and LAI is adopted in simplified growing production day time-stepping scheme (SGPD-TS) model (Xin et al., 2019). The SGPD-TS model, however, does not use the leaf biomass growth concept, rather

it establishes a linear relationship between steady-state GPP and LAI. In this way, GPP is used as a proxy of LAI, utilising a conversion ratio when maximum GPP has been reached. However, it has been shown that modelled GPP saturates at high LAI values (Lee et al., 2019). This may potentially introduce uncertainty when calculating the conversion ratio to simulate LAI. Another more general challenging aspect for these models is the specification of effective model parameters such that they can seamlessly operate at different scales and locations. Previous applications often have been limited to a calibration site

(Francés et al., 2007); but they need to be thoroughly cross-validated for their applicability across a diverse range of climatic conditions. The overarching aim of this study are to propose a parsimonious model (i) to simulate daily dynamics of GPP and LAI over deciduous broad-leaved forest at a medium level of complexity (ii) also suitable for integration in existing hydrologic and ecologic models. We simulate processes related to the carbon cycle in the canopy at a forest stand of undetermined size, utilizing the LUE approach with implementation of a phenology submodel. The parsimonious approach and level of model

complexity are adapted based on readily available observational datasets across eddy flux tower stations. We apply a global sensitivity analysis to investigate model parameters' sensitivity to the model's output variables (i.e., GPP and LAI). Finally, we assess the generality and robustness of the underlying model parameterizations and demonstrate the model applicability over different sites conducting a cross-location transferability experiment.

## 2  Methodology

### 2.1  Model overview

The PCM model developed and presented in this study aims at providing a parsimonious representation of daily development of biomass of leaf (Bl) coupled to simulated gross primary productivity (GPP) over deciduous broad-leaved forest (DBF) ecosystems. Analogous to most of the LUE models treating the entire vegetation canopy as a big extended leaf (Guan et al., 2021), the PCM operates over forest stand scale and adapts parameters mainly from a biome properties look-up table (BPLUT)

(Running et al., 2000). Parameters such as specific leaf area index (SLA) in PCM represent an effective community-weighted parameters. Figure 1 shows a schematic representation of the PCM structure including carbon fluxes/stocks and interconnected processes related to plant canopy for DBF biomes. We focus on simulating Bl, which is related to LAI via the specific leaf area index parameter. The simulated LAI is, in turn, used in the calculation of the GPP.

PCM uses a daily time step during which it simulates the processes of carbon uptake, leaf respiration, carbon allocation, and

carbon decay from the leaf pool (canopy) using a mass balance equation (Istanbulluoglu et al., 2012; Yue and Unger, 2015; Pasquato et al., 2015; Melton and Arora, 2016; Ruiz-Perez et al., 2017). The main governing equation to simulate the daily development of GPP(t) and Bl(t) is:

$$\frac{d\mathrm{Bl(t)}}{dt} = \big(\mathrm{GPP(t)} - \mathrm{R_e(t)}\big)\lambda(\mathrm{t}) - \mathrm{D(t)} \tag{1}$$





where Bl(t) is leaf biomass, GPP(t) is gross primary productivity, $R_e$(t) is leaf respiration, $\lambda$(t) is carbon allocation coefficient and D(t) is leaf decay components at day t. All terms on the right hand side are calculated in the modules of the PCM. The LAI (related to Bl(t) in Eq. 1) is defined as:

$$\text{LAI(t)} = \text{Bl(t)} \cdot SLA \cdot f_{cov} \tag{2}$$

where $SLA$ is specific leaf area index, and $f_{cov}$ is the vegetation fractional coverage. In the following sections, the modeling approaches implemented for each submodel component are described in detail. A summary of the model inputs and underlying parameters is provided in Tables 2 and 3, respectively.

### 2.1.1   Gross Primary Productivity

The theoretical soundness and practical convenience of the LUE concept in estimating terrestrial GPP has been the main core of several model developments (Monteith, 1972; Wei et al., 2017; Running et al., 2000; Arora, 2002; Schaefer et al., 2012; Zhang et al., 2015) at the regional and global scales (Potter et al., 1993; Yuan et al., 2007; Xiao et al., 2004; Running et al., 2000). In this study, we likewise utilize the LUE approach, which theoretically relies on the concept of interception of photosynthetically active radiation by plant leaves and converting it into biomass through energy to biomass efficiency factor (i.e. LUE factor). As expressed in Eq. 1, the PCM simulation starts with assimilation of the carbon flux (GPP) by leaf component. The GPP flux (Eq. 3) is estimated as a product of incident photosynthetically active radiation (PAR), by fPAR, which is a fraction of PAR being absorbed by plant leaf, and an LUE factor, multiplied by a modifier factor when environmental constraints present ($\epsilon$).

$$\text{GPP(t)} = LUE \cdot \epsilon(t) \cdot \text{PAR(t)} \cdot \text{fPAR(t)} \tag{3}$$

where $LUE$ is biome-specific unstressed (or maximum) vegetation light use efficiency parameter. $fPAR$ is calculated as following (Ruimy et al., 1999; Xiao et al., 2004; Wu, 2012; Yuan et al., 2007):

$$\text{fPAR(t)} = c \cdot \left(1 - e^{-(k \cdot \text{LAI(t)})}\right) \tag{4}$$

where $c$ refers to maximum absorption at full light interception in deciduous broad-leaved forest biomes (Monsi and Saeki, 1953; Ruimy et al., 1994) and $k$ is the light extinction coefficient parameter.

$\epsilon$ (Eq. 4) is an overall and integrated modifier that corresponds to environmental stress factors. The overall modifier factor diminishes light use efficiency of vegetation from its potential value during unfavorable environmental conditions (Potter et al., 1993). These unfavorable conditions include for example high and/or low temperature fT, water availability fSM, and elevated vapor pressure deficit fVPD stress factors (Zhang et al., 2015; Pasquato et al., 2015).

In general, calculation of $\epsilon$ across different LUE models can be expressed either in minimum (Eq. 5) or multiplicative (Eq. 6) approaches to integrate different environmental stress factors. On the one hand, models such as Eddy Covariance-Light Use Efficiency (EC-LUE; (Yuan et al., 2007)) uses Liebig law of minimum stress that emphasise the most limiting resource to constrain GPP (Eq.5). On the other hand models such as Carnegie-Ames-Stanford Approach (CASA; (Potter et al., 1993)) and





Vegetation Photosynthesis Model (VPM; (Xiao et al., 2004)) follow a multiplicative approach of stresses (Eq.6). In the present
study, we opt for the first approach to integrate different stress factors and to calculate the $\epsilon$.

The first approach (minimum) is expressed as follows (Running et al., 2000; Sitch et al., 2003; Prince and Goward, 1995).

$$\epsilon(t) = \min(\text{fT}(t), \text{fVPD}(t), \text{fSM}(t)) \tag{5}$$

The second approach can be written in a multiplicative way as:

$$\epsilon(t) = \text{fT}(t) \cdot \text{fVPD}(t) \cdot \text{fSM}(t) \tag{6}$$

The individual stress factors are dimensionless scalars ranging between zero (full stress) and one (no stress), and are intro-
duced in more detail in the following section.

### 2.1.2 Environmental constrains and GPP

**I) Temperature stress factor (fT)**: The first reduction factor, fT, on GPP due to air temperature is calculated by including two
factors corresponding to low temperature $\rho_l$ (cold) and high temperature $\rho_h$ (heat) stress effects (Eqs. 7,8,9) (Sitch et al., 2003;
Fischer et al., 2016; Rödig et al., 2017).

$$\text{fT}(t) = \rho_\text{l}(t) \cdot \rho_\text{h}(t) \tag{7}$$

The stress induced by the cold stress factor ($\rho_l$) can be calculated as:

$$\rho_\text{l}(t) = (1 + e^{k_0 \cdot (k_1 - T(t))})^{-1}, \tag{8}$$

where,

$k_0 = \frac{2\ln(0.01/0.99)}{(T_{low} - T_{cold})}$, $k_1 = 0.5(T_{low} + T_{cold})$

The heat stress factor is calculated as:

$$\rho_\text{h}(t) = 1 - 0.01 \cdot e^{k_2 \cdot (T(t) - T_{hot})}, \tag{9}$$

$k_2 = \frac{\ln(0.99/0.01)}{(T_{high} - T_{hot})}$


where $T(t)$ is daily mean air temperature, $T_{low}$ and $T_{high}$ are DBF biome-specific parameters representing high and low
temperature limits for $CO_2$ assimilation, respectively. $T_{hot}$ and $T_{cold}$ are the monthly mean air temperature of the warmest
and coldest months, respectively, that a DBF biome can cope with, respectively (Boons-Prins, 2010; Bohn et al., 2014; Fischer
et al., 2016; Rödig et al., 2017).

**II) Vapour Pressure Deficit stress factor (fVPD)**: The canopy photosynthesis rate is strongly related to changes of vapour
pressure deficit (VPD) (Konings et al., 2017; Xin et al., 2019), as photosynthesis declines due to stomata closure (Yuan et al.,
2019) when atmospheric VPD increases. It can be modelled as follows in Eq. 10 (Jolly et al., 2005):

$$\text{fVPD}(t) = \max\left(\min\left(1 - \frac{VPD(t) - v_{min}}{v_{max} - v_{min}}, 1\right), 0\right) \tag{10}$$





where $VPD(t)$ is daily vapour pressure deficit, $v_{min}$ and $v_{max}$ denote lower and upper thresholds for photosynthetic activi-
ties, respectively. The fVPD value of one indicates no stress on GPP, whereas there is full stress when the fVPD becomes zero;
values between zero and one result in partial and linear reduction on the GPP.

**III) Soil Moisture stress factor (fSM)**: In general, the impact of soil water deficit on photosynthesis in vegetation models is
represented as a generic soil moisture stress function using either modeled or field observation soil moisture content (Cox et al.,
1999; Granier et al., 2000; Fischer et al., 2016). Here, we use field observations from different vertical soil profiles including
volumetric soil moisture content and soil textural properties (wherever available) to calculate the soil moisture stress factor,
fSM.

Essentially, the soil moisture influence on plant productivity depends not only on soil moisture over the entire profile but also
on the available soil water to the plant roots. Therefore, to estimate the availability of water to plants, the characteristics of the
root system, including rooting depths and its distribution at different soil depths, are essential factors to be considered (Ostle
et al., 2009). Thus, we include plant rooting distribution in our analysis, following Jackson et al. (1996), to take into account
the root fraction at different soil depths, and weight the soil moisture content layer-wise according to the present fraction of
roots in that layer. In doing so, we calculated cumulative root fraction ($Rc_i$) from the surface to a certain depth (d) in the soil
profile for each layer ($i$) using the biome specific parameter, $\beta$ as follows (Eq. 11) (Jackson et al., 1996):

$$Rc_i = 1 - \beta^{d_i} \tag{11}$$

Then, using the cumulative root fraction up to each layer, the root fractions in each layer $Ri_i$ are estimated and then multiplied
with the corresponding observed soil moisture content of that layer to calculate the soil moisture contribution from each layer
individually. Later, by summing up the soil moisture contributions from all individual layers ($\theta_i$), a daily effective soil moisture
content, $\theta(t)$, over the soil column is obtained (Eq. 12-14).

$$Ri_i = Rc_i - Rc_{i-1} \tag{12}$$

$$\theta_i = \theta_i \cdot Ri_i \tag{13}$$

$$\theta(t) = \Sigma(\theta_i) \tag{14}$$

Similarly to other stress terms, the soil moisture stress factor varies between 0 and 1; and is quantified as follows (Eq. 15).

$$fSM(t) = \max\left(\min\left(\frac{\theta(t) - \theta_r}{\theta_{MSW} - \theta_r}, 1\right), 0\right) \tag{15}$$

where $\theta(t)$ is daily effective soil moisture, $\theta_r$ and $\theta_{MSW}$ are water storage corresponding to the permanent wilting point
and the critical point below which transpiration is limited, respectively. $\theta_{MSW}$, representing minimum soil water content for
unstressed photosynthesis (Hartge, 1980; Granier et al., 1999; Fischer et al., 2014), is calculated as follows:

$$\theta_{MSW} = \theta_r + scw \cdot (\theta_s - \theta_r) \tag{16}$$


where $\theta_s$ is soil water content at field capacity, scw is a constant threshold commonly set at 0.4, and a calibration parameter in PCM. scw is a physiological threshold defined as critical relative soil water content at which tree transpiration begins to decrease Granier et al. (1999). According to Granier et al. (1999); Fischer et al. (2016) the scw value does not vary significantly between soil and plant species and can be considered as a constant value. The $\theta_r$ and $\theta_s$ correspond to soil matric potentials of -1.5 and -0.033 MPa, respectively.

When the daily effective soil moisture content is above a minimum soil water content ($\theta_{MSW}$; Eq. 16), there is no stress to limit photosynthesis, while below the $\theta_{MSW}$ point, there is a linear increase in stress as water content decreases until $\theta_r$ is reached. At this point, the soil water stress factor becomes zero with full limitation on photosynthesis and GPP (Harper et al., 2021).

### 2.1.3   Canopy respiration

To allow estimation of daily changes in carbon in the leaf pool (Eq. 1), the release of carbon to the atmosphere from leaf respiration ($R_e$) has to be calculated. This flux is part of gained carbon (i.e., GPP) consumed for self-maintenance requirements in the leaf pool. In fact, canopy net primary productivity ($NPP_{canopy}$), which is net available carbon ready to be allocated among different plant pools, is the sum of photosynthetically carbon uptake by plants (GPP) reduced by carbon loss via leaf respiration ($R_e$) (Pasquato et al., 2015; Running et al., 2000; Melton and Arora, 2016).

We use the well-established modified Arrhenius equation (Eq. 17) (Lloyd and Taylor, 1994; Sitch et al., 2003; Perez, 2016) to calculate the leaf respiration. The $R_e$ flux is a function of air temperature, carbon mass of leaf pool, and a tissue-specific carbon to nitrogen ratio, given as:

$$R_e(t) = \frac{rr \cdot Bl(t)}{CNr} \cdot e^{p_1 \cdot \left( \frac{1}{p_2} - \frac{1}{T(t)+p_3} \right)}  \tag{17}$$

where $rr$ represents the leaf respiration rate, Bl the carbon mass of leaf pool (leaf biomass), $p_1$, $p_2$, $p_3$ are parameters in the
Arrhenius equation, $CNr$ is carbon to nitrogen ratio in leaves, and $T$ is daily mean air temperature.

### 2.1.4   Vegetation phenology module

We incorporated a phenology submodel into our model using the approach defined in Yue and Unger (2015). This submodel calculates temperature-dependent phenological factors for spring and autumn, $f_{ST}$ and $f_{AT}$ respectively. These factors range from 0 to 1 throughout the year, to determine the timing of spring budburst (once the spring temperature dependent factor sets up
to increase above zero), maturity (when the spring temperature-dependent factor approaches to 1), autumn senescence (once the product of autumn temperature-dependent and photo-period factors start off to decrease below 1), and dormancy phenophases (once the product of autumn temperature-dependent and photo-period factors approach zero). The second phenological factor in the autumn and dormancy phenology is photo-period ($f_{dl}$) factor and depends on day length. The photo-period factor together with the temperature-dependent factor regulate the leaf senescence. The phenology submodel determines the above-mentioned
four phenological transition dates on which a simple allocation of assimilated carbon to the leaf pool is based. Below, we provide details of each phenological factor and events.





**I) Spring phenology** ($f_{SP}$): The growing season starts with the budburst day, which is the beginning of canopy development and the time when green tips of leaf show up. It is estimated using a temperature-dependent phenological factor $f_{ST}$ as follows (Eq. 18):

$$f_{ST} = \begin{cases} \min\left(1, \frac{GDD - G_b}{L_g}\right) & GDD \geq G_b \\ 0 & \text{otherwise} \end{cases} \tag{18}$$

where GDD is growing degree day, $Gb$ is budburst threshold value, $L_g$ is a parameter for growing length in degree day. The accumulation of growing degree day (GDD) (Eq. 19) from winter solstice day is calculated as below:

$$GDD = \sum_{i=1}^{n} \max(T_{10} - T_b, 0) \tag{19}$$

where $T_{10}$ is 10-day average air temperature, $T_b$ is base temperature for the budburst (5°C).

$G_b$ in the estimation of $f_{ST}$ (Eq. 18) is a threshold value for budburst to occur and is calculated as follows:

$$G_b = a + b \cdot e^{(r \cdot NCD)} \tag{20}$$

where $a$, $b$, and $r$ are parameters for the budburst threshold. NCD is counted as number of chill days between the previous winter solstice day and the beginning of the successive year. Given the GDD and $G_b$ estimates, temperature-dependent phenological factor ($f_{ST}$) is then applied to calculate the spring phenology ($f_{SP}$) (Eq. 21).

$$f_{SP} = f_{ST} \tag{21}$$

**II) Autumn phenology** ($f_{AP}$): For the autumn phenology the product of two phenological factors, temperature $f_{AT}$ and photo-period $f_{dl}$ factors, is considered to estimate timing of senescence and dormancy. The autumn temperature-dependent factor, $f_{AT}$, (Eq. 22), is obtained as follows:

$$f_{AT} = \begin{cases} \max(0, 1 + \frac{(FDD - F_s)}{L_f}) & FDD \leq F_s \\ 1 & \text{otherwise} \end{cases} \tag{22}$$

where $F_s$ is a threshold in degree day for leaf fall, and $L_f$ is a threshold in degree day for the duration and length of the leaf falling period (more detail can be found in Yue and Unger (2015)). FDD (Eq. 23) is an accumulative falling degree day from summer solstice day which is known as a cumulative cold summation method (Yue and Unger, 2015) and it can be calculated as:

$$FDD = \sum_{i=1}^{m} \min(T_{10day} - T_s, 0) \tag{23}$$

where $T_{10day}$ is 10-day average air temperature, $T_s$ is base temperature for leaf fall at 20°C.

In addition to temperature factor $f_{AT}$, autumn senescence timing is regulated via photo-period factor $f_{dl}$, which is calculated





based on day length (dl) period, together with lower $(dl_{min})$ and upper $(dl_{max})$ limits of day length affecting leaf fall as in Eq. 24.

$$f_{dl} = \begin{cases} \max(0, \frac{dl-dl_{min}}{dl_{max}-dl_{min}}) & dl \leq dl_{max} \\ 1 & \text{otherwise} \end{cases} \tag{24}$$

where $dl$ is the day length in minutes. $dl_{min}$ and $dl_{max}$ are the lower and upper limits of day length for the period of leaf fall, respectively. The autumn phenology ($f_{AP}$) is finally calculated as a product of $f_{AT}$ and $f_{dl}$ (Eq. 25):

$$f_{AP} = f_{AT} \cdot f_{dl} \tag{25}$$

The predicted phenological transition dates from spring $f_{SP}$ and autumn $f_{AP}$ phenology factors determine the budburst-maturity and senescence-dormancy events, respectively. Based on this information, a fractional allocation to and decay from 280 the leaf pool is considered (as detailed below).

### 2.1.5 Carbon allocation to and decay from the leaf pool

The next step of the carbon pathway in Eq. 1 is allocation to and decay of assimilated carbon from the leaf pool. The leaf biomass state variable (Bl) in Eq. 1 is updated at a daily time-step, based on changes in gain and loss of carbon in the leaf pool. The allocation and decay processes are both key physiological processes in the vegetation models to govern the partitioning of 285 growth among different plant carbon pools and are critical determinants of plant productivity (Haverd et al., 2016; Xia et al., 2017). There are two widely used allocation schemes used in vegetation models based on: (1) fixed allocation coefficients, and (2) allocation driven by allometric constraints. The first scheme uses a fixed allocation ratio to individual plant's carbon pools (e.g. used in CASA (Friedlingstein et al., 1999) or BIOM-BGC(Hidy et al., 2016)). In this scheme, the allocation ratio is constant within different plant development stages. In the second scheme, a fraction of carbon is allocated in such a way that 290 it satisfies allometric relationships that exist between various plant compartments (Malhi et al., 2011; Gim et al., 2017). In the case of allocation to leaf, the allometric relationship is based on the relative mass of canopy – so-called maximum $L_b$ – that a plant can support with a certain stem mass and height. We adopted an allocation scheme that mainly depends on an updated daily carbon status of the leaf pool. We use the maximum values of balanced LAI supported by the system (Eq. 26) based on a previous study conducted by Fleischer et al. (2013). Instead of considering it as a fixed value, we vary $L_b$ within a range of 295 $\pm 1 m^2/m^2$, and consider it as one of the model parameters.

$$\lambda(t) = 1 - \frac{LAI(t)}{L_b} \tag{26}$$

where $\lambda(t)$ is the carbon allocation ratio to the leaf pool and $L_b$ is the maximum LAI that can be supported by plants.

Provided with the identified major phenological transition dates from the phenology submodel – i.e., budburst, maturity or steady growth, senescence, and dormancy – the calendar year is accordingly divided into four main stages. During the early 300 growing season, once the climate condition becomes favourable to plant growth and budburst occurs, carbon allocation to leaf, $\lambda$ (Eq. 26), is relatively a large fraction. This means that the largest part of carbon will be partitioned towards leaf and is being





used for growth during the early growing season (Gim et al., 2017). Given the value for balanced LAI supported by the system (Fleischer et al., 2013), the carbon allocation slowly decreases with an increase in LAI until the leaf mass reaches that balanced LAI. As soon as the canopy approaches a full leaf state (i.e. maturity phenophase), the carbon allocation ratio to the leaf is held

at its minimum – a small portion is used for maintenance respiration during this steady growth stage. We set the leaf allocation ratio during the maturity phase to a value of 5% from the assimilated carbon, following the recent version of the Noah-MP model's leaf allocation scheme (Gim et al., 2017).

After the steady growth and maturity phase, the leaf senescence phase approaches and the leaf-loss processes start to play the main role in moderating the mass-balance of canopy and the corresponding LAI seasonality. The loss of carbon via the leaf

fall in PCM is simulated based on the calculated senescence and dormancy transition dates via the phenology submodel, such that when the simulation time-step approaches to the senescence date, the model linearly decreases the leaf biomass until the leaf biomass reaches to nearly zero at the beginning of the dormancy phase.

Concerning the leaf loss processes, PCM also accounts for the leaf losses due to cold stress ($O_C$) (Eq. 27), drought stress ($O_D$) (Eq. 29), and normal loss of the leaf ($O_N$) (Eq. 30) following schemes of the CLASSIC model (Melton and Arora, 2016).

The leaf loss due to the cold stress is given by:

$$O_C(t) = O_{Cmax} \cdot (Cs(t))^3 \tag{27}$$

where, $O_{Cmax}$ is the maximum leaf loss rate parameter and $Cs$ is a cold stress factor value. The cold stress factor (Eq. 28), ranging between 1 (full stress) and 0 (no stress), is calculated as:

$$Cs(t) = \begin{cases} 1 & T(t) \leq (T_c - 5) \\ 1 - \frac{T(t) - (T_c - 5)}{5} & (T_c - 5) < T(t) < T_c \\ 0 & T_c \geq T(t) \end{cases} \tag{28}$$

where $T(t)$ is air temperature and $T_c$ is a biome specific temperature threshold below which leaf damage is expected.

Similar to the $O_C$, the leaf loss rates due to drought stress $O_D$ (Eq. 29) is calculated using the fSM stress factor (through the soil moisture stress submodel) and a $O_{Cmax}$ maximum leaf loss rate parameter associated with the drought stress.

$$O_D(t) = O_{Dmax} \cdot (1 - fSM(t))^3 \tag{29}$$

The third leaf loss term represents the loss rates due to a Normal decay $O_N$ driven by biome specific leaf lifespan ($\tau = 1$ for

DBF in Eq. 30) given by:

$$O_N(t) = 1/(365 \cdot \tau) \tag{30}$$

Finally, the total decay of leaves $D(t)$ consists of contributions from all individual losses (Melton and Arora, 2016); and can be given as follows (Eq. 31):

$$D(t) = Bl(t) \cdot \left(1 - e^{-(O_C(t) + O_D(t) + O_N(t))}\right) \tag{31}$$





where $O_C$, $O_D$, and $O_N$ are the leaf loss rates due to cold stress, drought stress, and normal decay, respectively.

In summary, the proposed PCM model comprises the submodels mentioned above in a hierarchical chain, starting with the carbon uptake via the initial leaf biomass state variable and continues with daily partitioning of that assimilated carbon together with daily decay from leaf compartment to calculate the leaf biomass production increment. This biomass increment is later added up to the state variable from the previous time step to update the leaf biomass for the current time step. Finally, to update

the LAI that is required for the GPP estimation over the next time step, the current leaf biomass is converted to LAI according to Eq. 2.

## 2.2    Model set-up and experimental design

### 2.2.1    Study sites and datasets

This study focuses on deciduous broad-leaved forests biome type and We selected tower sites distributed over Europe and

North America to ensure a representative spatial coverage. Sites were excluded if data of fewer than five years were available. We further screened out the data at each site to the years with minimal gap in input data, in particular, photosynthetic photon flux density variable and its associated frequent and long missing data at some sites. Applying the above criteria, nine sites with varying temporal coverage were retained for the analyses (Fig. 2). The general site information is presented in Table 1. Daily flux and meteorological forcing data are from ecosystem stations available from the free fair-use

FLUXNET2015 Tier 1 global collection database (https://fluxnet.org/data/download-data/, last access: June 2021) (Pastorello et al., 2020). The input data required to drive the PCM comprises: air temperature (T), photosynthetic active radiation (PAR) (i.e. converted from PPFD in $\mu mol\ m^{-2}\ s^{-1}$) and vapor pressure deficit (VPD) (Table 2). The tower-based GPP estimations, GPP_NT_VUT_REF from the FLUXNET2015 dataset are used for model calibration. We used the first year of the time series as a warm-up period, during which the chilling days and thermal requirement in phenology submodel are counted.

Optional variables to establish the model include soil moisture (SM) and soil textural properties, are required to simulate the soil moisture stress development. However, we investigate the soil moisture stress impact only at the Hohes Holz (DE-HoH) site in Germany with soil moisture data available up to 80 cm depth. In regard to calculating the soil moisture stress in PCM, a pedotransfer function following Zacharias and Wessolek (2007) is implemented to estimate site-specific $\theta_s$ and $\theta_r$ values. This (pedotransfer) submodel receives soil textural properties (sand, clay contents, and bulk density) obtained from

field observations of spatially distributed soil profiles as input. It provides the required field capacity ($\theta_s$) and permanent wilting point ($\theta_r$) to calculate $\theta_{MSW}$ and the corresponding soil moisture stress term fSM in the calculation of $\epsilon$ (Eq. 5). To maintain the consistency with the vertically weighted soil moisture, $\theta_s$ and $\theta_r$ are estimated as weighted average values of individual layer-specific $\theta_s$ and $\theta_r$ taking the respective root fractions as a weighting factor. Other required parameters in the model related to different processes, are listed in Table 3. The LAI field measurements were collected via personal commu-

nication to site contact persons; and based on the responses a subset of 4 sites (DE-HoH, DE-Hai, US-MMS, and US-Ha1 (https://harvardforest1.fas.harvard.edu/exist/apps/datasets/showData.html?id=hf069, last access: 05 January 2022)) are used to evaluate the modeled LAI. The observation-based LAI measurements are obtained using common procedures of LAI-2000





instrument (Gower and Norman, 1991) and fisheye technique (Bonhomme, R. and Chartier, P., 1972). These two methods are considered as the closest methods yielding similar values among other techniques and, therefore, provide consistent measure-

ments(Ariza-Carricondo et al., 2019).

### 2.2.2   Model structure and set up

The impact of water availability on the canopy photosynthesis (i.e., soil water deficit and atmospheric water deficit), in vegetation models is structured in two ways; individually or in combination with each other. In some models, water stress is quantified as an overall stress from both atmosphere and soil-moisture (GLO-PEM; Prince and Goward, 1995), (Biom-BGC; Hidy et al.,

2016), while some other models account for the water stress due to either the atmospheric drought (CASA; Potter et al., 1993), MOD17 algorithm (Running et al., 2000)) or soil moisture drought (EC-LUE; Yuan et al., 2007). In order to determine, how stress should be represented in the final version of PCM, we conduct two sets of preliminary model experiments to examine: (1) whether inclusion of fSM, additionally to the other stress factors affects the results, and (2) the effect of alternative integration approaches (i.e. Liebig law and multiplicative approaches, see Section 2.1.1) on simulated GPP over the DE-HoH site during

the drought 2018. Since the best model skill of the PCM was achieved, when incorporating all stress factors (fT, fVPD, and fSM) in the calculation of the overall environmental stress; and using the minimum integration approach (Eq. 6), this structure was selected for the final setup (see Figures in Supplement, Figure S1 and Figure S2). With regard to specific considerations in LAI simulations, the model starts with the simulation using a fixed initial LAI state variable to begin the carbon assimilation once weather conditions become more favourable for plant growth. Following the CABLE model parameterizations  (Li et al.,

2018), we set the initial LAI value to 0.35. We also consider a local maximum LAI (so-called $L_b$ in this study), obtained from reported values in literature (Fleischer et al., 2013), that individual mature forest can sustain at canopy closure. However, the local maximum LAI is, later in the calibration step, allowed to vary within $\pm 1\mathrm{m}^2\mathrm{m}^{-2}$ of the reported value. The $L_b$ constrains the simulated LAI up to the reported value at each site across years.

### 2.2.3   Global sensitivity analysis

Despite the simplicity of parsimonious models, assessing model robustness remains a fundamental step when building and developing a model. One of the powerful and invaluable tools for robustness assessment is global sensitivity analysis (GSA) to test the underlying model parameterizations and inform about sensitive model parameters for the subsequent parameter inference. In general, the GSA can be performed to understand the influence of parameters perturbations on modeled simulations and to determine the informative parameters that contribute the most to an output behavior (Iooss and Lemaître, 2014; Cuntz

et al., 2016; Rakovec et al., 2014). In this study during the GSA, the parameters vary over boundaries reported in literature's. In case there were no bounds available for some parameters (e.g., phenological parameters from Yue and Unger (2015)), we varied them at $\pm$ 20% level of their default values. We utilize the Sobol' variance-based sensitivity method (Saltelli et al., 1999) with Sobol2002 formula (Saltelli, 2002), in which decomposition of the output variance is performed in terms of Sobol' indices. The Sobol' First order index (Si) and total-order Sobol' index (ST) express the share of output variance associated

with a given parameter $i$ and the share of output variance where all parameters are varied except the parameter $i$, respectively.



These indices range between 0 to 1; with zero value indicating that the model output is entirely insensitive to the respective parameter changes. The closer the value to 1, the more important and sensitive the respective parameter is. Generally, the model parameters deem sensitive, if the sensitivity index is above a certain threshold value. Here in this study, we report the total-order Sobol' index and set the selection threshold at 1% (Tang et al., 2007), meaning that if the variation of a given

parameter contributes to a change greater than 1%, then that parameter is recognized as an informative parameter. In contrast, non-informative parameters are reported as parameters with Sobol' indices below 1%. Given the focus of the present study on two main output variables (i.e. GPP and LAI), we use the time mean for both outputs over the entire period for the sensitivity analysis at each study site. However, the results are expected to differ not only according to the site and selected target output but also between the individual years if a specific year is of interest to be investigated (Göhler et al., 2013; Hou et al., 2012). To

conduct the sensitivity analysis, we opt to choose all coefficients in the empirical equations as adjustable parameters (Table 3). It helps to explore the model sensitivities of often hidden parameters to properly constrain the model (Cuntz et al., 2016). Overall, we apply the global sensitivity analysis in all study sites for the common 29 parameters and analyse the sensitivity of the soil moisture stress parameters together with other parameters only for the DE-HoH site at which representative soil moisture data at different depths, down to 80 cm into the soil, was available. Given the importance of the number of model

evaluations required to conduct the Sobol' sensitivity analysis (Nossent and Bauwens, 2012) and the stability of sensitivity indices, we also check the convergence of the Sobol' indices through a visual assessment of diagnostic plots.

### 2.2.4 Parameter estimation

Based on the results of sensitivity analysis, informative and non-informative parameters are identified. Later, we fixed the non-informative parameters to their corresponding reported values in literature (see Table 3 for details) and the remaining

informative parameters are inferred using a Monte Carlo approach (Kuczera and Parent, 1998). The parameters were calibrated against the GPP_NT_VUT_REF time series from the corresponding flux tower measurements (global Fluxnet Tier1 network accessed on 13 February 2021) (Pastorello et al., 2020). It is important to note that besides the maximum LAI value we did not use LAI field observations in the calibration process as LAI is not readily available from the FLUXNET dataset. Instead some LAI observations (obtained from site contacts) were used in the model verification step. The first year of the dataset is

considered a spin-up period. The rest of the time-series are divided into two sub-periods. The first half is used for the calibration phase, and the remaining years to independently evaluate the model performance (i.e., over the out-of-calibration set). A total of 10 000 parameter sets was sampled from their a priori defined ranges (Table 3) in each study site to estimate the parameters and simulate the GPP flux and LAI. Model performance was quantified using a group of performance metrics, including Kling-Gupta efficiency (KGE) (Gupta et al., 2009), Root Mean Square Error (RMSE), and coefficient of determination ($R^2$).

We selected an ensemble of informative model runs that simultaneously lie within the top 5% of all the performance metrics.

### 2.2.5 Site-specific verification and model generalization

The second half of the GPP time series at each study site was used for the model verification step. In addition to the at-site verification, it is also equally important to consider the generality of the model structure including underlying model





parameterizations. To this end, we considered an independent (spatial) verification approach – so called cross-validation – for
assessing the robustness of model parameterizations beyond the conditions during which they were calibrated. The relevance of
the cross-validation to the modeling framework, is that transferable models can be used beyond the spatial and temporal limits
of their underlying data, especially in the face of pervasive scarcity of observational data to constrain model parameterizations
(Yates et al., 2018). Therefore, as the next step in our modeling framework, and after performing the site-specific calibration
and verification, a cross-validation of the model is conducted to come up with a compromise solution (here parameter set)
applicable across the study sites, following the approach of Zink et al. (2016). In doing so, the behavioral parameter sets found
from on-site calibration for each study site are grouped together as one unique set of all possible behavioral parameters. Then
the model is run using all possible parameter sets and the respective performance metric (i.e., KGE) for each parameter set
at each investigated site is estimated. After that, the mean values of KGEs corresponding to each parameter set over all study
sites are calculated. Finally, a set of parameters associated with the highest mean KGE score is recognised as a compromise
solution. Here the goal of this analysis is to investigate the generality of the underlying model structure, and to allow inference
of a common (compromise) set of model parameters for the PCM for a broader applicability (i.e., beyond the calibration sites).

## 3 Results and Discussion

In the following, we first show and discuss findings from the global sensitivity analysis and site-specific parameter calibration.
This is followed by a discussion of the site-specific model performance. Finally, we present the results of a cross-validation to
test the generality of underlying model parameterizations. This also allows us to propose a standard set of PCM parameters for
locations where calibration is not possible.

### 3.1 Sensitivity analysis

Here, we explore the sensitivity of the output variables (i.e. GPP and LAI) to the model parameter variations using Sobol'
method at each study site. Although a direct comparison of PCM parameters sensitivities from this study with similar models
in other studies is difficult due to difference in model structures and representation of photosynthesis processes, one can gain
insights by comparative assessments among conducted studies. For instance, the light utilization in LUE-oriented GPP models
is based on photon absorption and photosynthetic efficiency of incident light (Frost-Christensen and Sand-Jensen, 1992).
Hence, one can compare the significance of the LUE parameter of our model with that of the quantum yield of photosynthesis
which is a measure of photosynthetic efficiency in the Farquhar equation (Farquhar et al., 1980) in several land surface models.
As it can be seen from Figure 3a (mean GPP) and b (mean LAI), different sensitive parameters are associated with the different
output variables. However, for the same output variable, all sites more or less share a similar informative set of parameters,
although the magnitudes differ. In the following, we show and discuss the sensitivity of the model outputs to different PCM
parameters.





### 3.1.1 Parameter sensitivity for GPP estimation

We first investigate the sensitivity of GPP output to the model parameters. Figure 3a shows the total-order Sobol' index of all parameters contributing to the GPP output. The boxes in Figure 3a indicate variation of the sensitivity of a given parameter across different sites. Only a small number of them have ultimate control on the simulated GPP out of the 34 model parameters (Figure 3a). This is in agreement with previous studies using LPJ-DGVM (Zaehle et al., 2005), BETHY (White et al., 2000), and BIOME-BGC (Knorr, 2000) models showing only a few of investigated parameters significantly influence the modelled

carbon fluxes outputs (including GPP).

The most sensitive parameter for the GPP estimates turned out to be the light use efficiency, LUE in (Eq. 3). This agrees with numerous other studies confirming that the light use efficiency is a significant parameter affecting GPP. For instance, Zaehle et al. (2005) conducted a probability-based sensitivity analysis using the LPJ-DGVM ecosystem model, utilizing Farquhar photosynthesis scheme, and found that carbon fluxes (including GPP) are highly sensitive to parameters related to photosynthesis

process, particularly the intrinsic quantum efficiency parameter (so called $\alpha_q$ in their model), which is related to the LUE in PCM. Similarly, Ma et al. (2020) using a GSA in the Flux-based Ecosystem Model and based on the Farquhar photosynthesis scheme, found canopy quantum efficiency of photon conversion among the most sensitive parameters with a strong influence on forest GPP. The multiplicative coefficient of canopy reflectance, C, and the light extinction coefficient, k, parameters in the fPAR formulation (Eq. 4) based on Lambert-Beer's law show also substantial sensitivities. Notably, these parameters are

typically fixed to constant values by default in the fPAR formulation, controlling PAR availability and utilization, in similar studies Xiao et al. (e.g. 2004); Xin et al. (e.g. 2019).

The next group of sensitive parameters are those involved in the imposed environmental stresses on GPP: I) The $v_{min}$ parameter (Eq. 10) exhibits some sensitivity and controls the impact of vapour pressure deficit stress on simulated GPP (fVPD). Balzarolo et al. (2019) also reported the impact of VPD variable in general on radiation use efficiency and on resultant GPP. II)

Next environmental factor constraining the GPP is soil moisture stress. Here, we identify $\beta$ (Eq. 11) and $\theta_r$ (Eq. 15) as sensitive parameters. We can only consider and discuss the soil moisture stress-related parameters in the DE-HoH site due to the lack of soil moisture data at other sites. The investigated sensitivity of fSM-related parameters are shown in the supplementary Figure S3. Similar findings of a pronounced impact of parameters controlling soil moisture availability (e.g., $\theta_r$ and $\beta$) on simulated GPP has been reported by Li et al. (2016) for the CABLE and JULES models. From a soil science perspective, the

$\theta_r$ is often a fixed value of soil water content corresponding to soil matric potential of 1500 kPa (Zhang and Han, 2019) and is typically not considered as a parameter. However, our result shows that the $\theta_r$ might not be considered as fixed. While the functional form of $\theta_r$ can be deduced based on pedo-transfer functions (Zacharias and Wessolek, 2007), empirical coefficients of such functions representing the linkages between $\theta_r$ and soil textural properties (e.g., sand, clay contents) can be regarded as model parameters (Samaniego et al., 2010; Kumar et al., 2013; Schweppe et al., 2021).

The SLA parameter (Eq. 2), as one of the structural parameters, is also a major contributor to the simulated GPP. Its sensitivity can be explained by the direct effect of SLA on LAI calculation (Eq. 2) through which the carbon assimilation (GPP) is eventually taking place (Eq. 4, 3). Arsenault et al. (2018) and Li et al. (2016) also reported the SLA parameter among very



sensitive model parameters, when simulating carbon fluxes (including GPP) in the Noah-MP and CABLE land surface models, respectively.

Finally, the last group of sensitive parameters in modeled GPP are those involved in the phenology submodel. The parameter $F_s$ (Eq. 21), determining the timing of leaf fall, appeared as a major informative parameter for all sites. Although, some parameters were only sensitive in some sites including those for the leaf budburst threshold- namely, b and r (Eq. 19). The b appeared sensitive only at DE-HoH and the parameter r is sensitive at CA-Oas and US-Oho. Generally, the implemented phenology submodel controls the plant active period and at the same time accounts for the impact of temperature factor on leaf

development and resultant GPP. This might be a reason why temperature-related parameters in the temperature stress factors (Eqs. 8 and 9) are not found to be informative in the sensitivity analysis. This is because temperature mainly controls the start and end of the growing season in the phenology submodel. This period indicates favourable condition for plant growth when the temperature stress is mostly not active. Therefore, corresponding parameters do not significantly influence the modelled GPP. In agreement with our results, Yuan et al. (2007) also reported little impact of environmental stresses due to temperature

on GPP during the growing season. Another interesting point emerging from Figure 3a is the insensitivity of GPP output to the LAI-balanced (maximum), $L_b$. This effect can also be seen in LAI simulation (e.g., at DE-HoH site) where a group of daily LAI at the maturity phase in Figure 7 lead to not much of the difference in the corresponding GPP outputs (i.e., in Figure 5. This is in agreement with the previous studies of Lee et al. (2019); Jung et al. (2007), which showed that GPP output saturates and becomes insensitive at LAI values above 4 $m^2\ m^{-2}$.

### 510   3.1.2   Parameter sensitivity for LAI estimation

We further explore the parameter sensitivity for LAI output similar to the GPP described above. In general, a similar set of sensitive parameters were identified for GPP and LAI outputs across sites (Figure 3b). However, some differences were also detected: parameters such as $L_b$, $f_{cov}$, $L_g$, $p_2$, and $p_3$ show substantial sensitivity, while the sensitivity to $v_{min}$ was almost negligible. Regarding the similarity of parameters between GPP and LAI, it is worth noting that the calculations of GPP and

LAI depend on each other since assimilated carbon (i.e.,GPP) is partly converted to leaf biomass from which the LAI is calculated, and used in turn for the GPP calculation in the next time step. Therefore, LAI output should roughly be sensitive to the same set of parameters as the GPP output. The result in Figure 3b shows that LUE, C, and k, directly involved in the GPP formulation, have considerable influence on the LAI output. These parameters, in particular the LUE, strongly control the assimilated carbon and consequently affect the modelled LAI.

Figure 3b also shows a major contribution of SLA (Eq. 2), $f_{cov}$ (Eq. 2), and $L_b$ (Eq. 24) to the LAI output. Similarly to the LUE for GPP, the SLA is central for the calculation of LAI (Eq. 2) and thus shows by far the largest sensitivity. Since the LAI output in the model depends on GPP, the studies reporting the SLA impact on GPP might apply for LAI output as well (Li et al., 2016; Arsenault et al., 2018). The $f_{cov}$ parameter represents the fractional vegetation coverage per unit area and is a critical parameter in calculating forest GPP (Ma et al., 2015). Ma et al. (2015) assumed 100% forest coverage in their calculation

of GPP, from which LAI was calculated. They showed how this inappropriate assumption (i.e., 100% forest coverage) can exaggerate the forest area when calculating forest GPP (and consequently the LAI) rather than considering a realistic coverage.





Here in the PCM, the $f_{cov}$ parameter is allowed to vary between 60% to 95% as an adjustable parameter (based on Fluxnet2015 Dataset description of percent coverage greater than 60% at DBF sites; http://sites.fluxdata.org/). We observe that fractional vegetation coverage substantially influences the simulation of LAI. In agreement with Ma et al. (2015), our result supports the importance of the fractional coverage ($f_{cov}$) as an important structural parameter in carbon cycle modelling studies. The $L_b$ parameter (Eq. 24), also exhibits a marked sensitivity for the LAI output (Figure 3b) because this parameter is a direct factor allowing the canopy to reach to its maximum. Next important contribution of parameters to the LAI output are those governing the leaf phenology in the phenology submodel, $L_g$ (Eq. 18), $F_s$ (Eq. 22), b (Eq. 20), r (Eq. 20) (Figure 3b). To the best of our knowledge, these parameters have not been studied elsewhere within a sensitivity analysis framework, and therefore we could not perform any comparative assessment. Parameters b and r contribute to the simulation of leaf budburst day, $F_s$ contributes to the identification of leaf fall day, and $L_g$ parameter influences the LAI output estimation through its influence on the length of the growing season. The $F_s$ parameter exhibits higher sensitivity and a larger between-site variation than other parameters (Figure 3b). This parameter represents the necessary amount of cold accumulation in degree day to trigger the leaf fall event. For instance, lower cold degree days accumulation would lead to an early leaf fall and leaf shedding. Therefore, the between-site variation of this parameter might not be surprising, given the differences in temperature and accumulated cold degree days among study sites.

Other additional parameters that showed sensitivity for the LAI output are $p_2$, and $p_3$ (Eq. 17). These parameters belong to the canopy respiration process in the modified Arrhenius equation (Eq. 17). They are typically considered as fixed parameters e.g., in TETIS-VEG model (Perez, 2016), in LPJ-ML model (Schaphoff et al., 2017), while here we varied these parameters within 20% of their nominal value. Notably, these parameters showed greater sensitivity for the LAI estimation than that of the GPP. It might partly be due to the reduced assimilated carbon (GPP) by canopy respiration which in turn might decrease the available carbon to be allocated to leaf biomass and affect the resultant LAI. Furthermore, the evaluation of Sobol' indices convergence (see Figure 4) showed relative stability of sensitivity indices at around 8 000 model evaluations.

## 3.2 Site specific model assessment

We conduct site-specific parameter estimation for all available eddy-covariance (EC) flux tower study sites (Figure 5). For this, only informative parameters identified in the sensitivity analysis are calibrated and the others are fixed (Table 3). For model parameter calibrations we used the first half of the available time series and the remaining years for verification (Table 1). Calibration and verification of the model are only performed for the GPP flux because direct LAI measurements are not available at all of the flux sites. Figure 5 shows the seven-day mean of simulated GPP for a set of ensemble members for each study site during both the calibration and verification periods. Since the different sites were operational at different times and some sites (e.g. DE-Hai) cover a relatively long time period, we show only five years of simulation at each site: the last three years of calibration and the first two years of verification periods (Figure 5). A complete simulation for each site during the entire available times series is provided in the Supplementary Figure S4. In addition, Table 4 summarizes the model performance in simulating GPP during calibration and verification periods at different sites. In general, the model achieved KGE values of above 0.65, RMSE of less than 2.5 $gCm^{-2}day^{-1}$, and $R^2$ values of above 0.65 over all study sites. We





compare the performance of our model to other modeling studies, whenever there is either an identical site to our study or a similar biome type (i.e., DBF) presented. To this end, our results are similar to Yue and Unger (2015) who found a high correlation of more than 0.8 and RMSE lower than 3 $g\ C\ m^{-2}\ d^{-1}$ for the GPP simulations at DBF forest sites in a global setting using the Yale Interactive terrestrial Biosphere model. Another study conducted by Asaadi et al. (2018) showed a quite

good model performance using the CLASS-CTEM land surface model (Melton and Arora, 2014) applied at US-Ha1(1998-2013) and US-MMs (1999-2006) flux tower sites, with $R^2$ value of 0.99 accompanying RMSE of 0.62, and $R^2$ value of 0.98 accompanying RMSE of 1.07 $g\ C\ m^{-2}\ d^{-1}$ at US-Ha1 and US-MMs, respectively. In a recent study, Holtmann et al. (2021) assessed the daily carbon fluxes over the DE-HoH forest during 2015-2017 using the FORMIND model (Fischer et al., 2016). They showed that the simulated and measured GPP correlates with an $R^2$ of 0.82 and RMSE of 9 $MgCha^{-1}a^{-1}$ equivalent to

2.46 $g\ C\ m^{-2}\ d^{-1}$ using FORMIND model.

Taken together, our model exhibits a reasonable validity and performs equally well in estimating temporal dynamics of GPP (Table 4) compared to other more complex land surface and biogeochemical models. The PCM shows skill in capturing GPP at most of the investigated sites; although it overestimates GPP at the IT-Ro1 during summer. IT-Ro1 is located in a Mediterranean climate exposed to dry summers (Vicca et al., 2016). The forest ecosystems in Mediterranean type climate are affected by water

limitation which can affect the GPP flux significantly (Cueva et al., 2021). The lack of soil moisture data probably contributed to the misrepresentation of GPP at this site. This is in agreement with previous studies that found similarly poor modeling performance across sites located in the Mediterranean climate in central Italy in dry summer periods when simulating GPP (Maselli et al., 2012; Chiesi et al., 2011; Fibbi et al., 2019). In addition, water limitation impact on GPP could be related to the irregular occurrence of extreme events (e.g., European-wide drought 2018). Such conditions were observed at DE-HoH

and DE-Hai sites, where the model overestimated GPP during late summer of 2018 coincided with Europe-wide drought 2018 (Buras et al., 2020). In the next step, we also examine the model's overall performance in reproducing GPP in terms of multi-year average of GPP at each site. Figure 6 shows that the model can generally explain the spatial variation of GPP with an $R^2$ as high as 0.90.

As an independent verification step, we evaluate the PCM simulations of LAI against field-measurements data at some study

sites where observational data were made available via personal contacts with site investigators. Figure 7 compares simulated values of LAI with their field measurements at four sites (US-MMS, US-Ha1, DE-Hai, and DE-HoH). In general, a good spatial and temporal consistency between the simulated LAI and the field-measurement LAI can be seen from the plots (Figure 7). The $R^2$ corresponding to the US-MMS, US-Ha1, DE-Hai, and DE-HoH sites are 0.90, 0.85, 0.78, and 0.90, respectively. Furthermore, the comparisons yield RMSE of 0.96, 1.58, 2.21, 1.4 $m^2m^{-2}$ to the US-MMS, US-Ha1, DE-Hai, and DE-HoH

sites, respectively. Table 5 summarizes the model performance in simulating LAI during a common period of observed and modeled data.

The simulated LAI captures reasonably well the observed LAI seasonality at almost all the sites. It demonstrates the capability of the model in capturing canopy status at different phenophases. However, there are some pronounced deviations between modelled and observed LAI at some sites (US-Ha1, DE-HoH) during the dormancy phase and autumn leaf decline period.

Given the deciduous nature of those ecosystems , it is likely that the higher winter values of field measurements compared





to simulated LAI reflects the presence of understory vegetation (Asaadi et al., 2018) or instrument's reading of present stand and/or dead leaf on trees after onset of leaf shedding. We also notice a slightly lagging phase in simulated LAI during the spring as compared to the field-measurements data, for instance at the DE-Hai site. Such discrepancy may be due to the lack of accounting for dependence of green-up rate on non-structural carbohydrate from previous years as a buffer to initiate leaf
onset (Asaadi et al., 2018), which is currently not represented in the PCM.

### 3.3 Spatial model verification and model generalization

Eventually, we conduct cross-validation of parameter transferability for all sites against tower-derived GPP data (Section 2.2.5). Figure 8 summarizes the results of this analysis, providing a comparison between the range of obtained Kling-Gupta efficiency performance metric (KGE) from on-site calibration and KGE obtained from a compromised solution. It can be seen that the
model with a compromise parameter set still shows a reasonable predictive skill (KGE > 0.5), while leaving space for skill improvement with a site-specific parameter ($\Delta\,KGE \approx 0.10$). The poorest performances are associated with the northernmost site DK-Sor and the Mediterranean IT-Ro1 site. The associated bias in those sites is likely related to GPP response to the maximum LUE parameter obtained from compromise solution applied over all the sites. As it was shown in the sensitivity analysis (see Section 3.1.1), the variation of GPP is predominantly driven by the LUE variation thus a constant fixed maximum
LUE across all sites might be a reason for the limited performance at the sites located in maximum latitude and water-limited regions. It has been shown that maximum LUE varies in different geographical locations (Jung et al., 2007), and this is in line with our on-site calibration result indicating largest maximum LUE at DK-Sor (northernmost site with a cold and moist climate) and lowest at IT-Ro1 (a relatively drier Mediterranean site) sites. Thus applying a compromise value for LUE at these two site would result in a bias in GPP estimation. Previous studies (Wang et al., 2010; Madani et al., 2014) showed a large
variation in maximum LUE not only between different biomes but also even within an individual biome and plant functional type. Concerning the large spatial variability of maximum LUE, several factors such as spatial heterogeneity of vegetation, canopy densities, ages, soil nutrition, leaf nutrient content have been mentioned in previous studies (Wang et al., 2010; Madani et al., 2014). Some methods such as spatially explicit estimation of optimum LUE (Madani et al., 2014) or introducing pixel-level maximum LUE (Wang et al., 2010) have been employed in satellite-based LUE models to overcome this shortcoming.
They argued that the assumption of a constant maximum LUE (i.e. based on standard MODIS-base GPP algorithm and a Biome Property Look-Up Table; Heinsch et al., 2003), needs to be reexamined so that spatial heterogeneity between individual plant functional types is represented. Therefore, the uncertainty introduced by the fixed maximum LUE may be reduced and ecosystem productivity modeling would be improved.

### 3.4 Limitation and opportunities

While the model performs well, in general, on simulating the GPP, some inconsistencies in the observed and modelled GPP across sites help to identify the model limitation and introduce future opportunities to improve the model performance. One of the mismatches is that the model lacks to adequately capture the observed decline in GPP during 2019 (Figure 5) at the DE-HoH. This may be related to a possible legacy impact of the drought year 2018 into the successive year 2019 (Buras et al.,





2020; Schuldt et al., 2020; Schnabel et al., 2021; Reichstein et al., 2013). Here we infer that the reduction in the tower GPP

in 2019 might be due to a change in the LUE parameter. Based on calibration from previous non-drought years, the obtained LUE value might lead the model to overestimate GPP in early 2019. Indeed, calibrating the model to the drought years of 2018 and 2019 yielded a lower LUE parameter (reduction of LUE value by 12%), which might support the possible legacy impact of last year drought on LUE parameter. Another possible explanation, alternatively or collectively to the plant legacy effect, would be variation/depletion of deep soil moisture storage (Jung et al., 2009). Since the model does not represent established

internal feedback for carrying over effect after extreme events (Reichstein et al., 2013) and only consider the soil moisture up to 80 cm depth, thus the current model version would not reflect on such a process and GPP is likely to be overestimated.

Another limitation in our simulation is a lack to account for a possible effect of diffuse light on GPP response in the current model structure. We observed the potential role of diffuse light on GPP during some mismatch periods between eddy flux tower and modelled GPP across some of the sites (e.g., DE-HoH year 2107, FR-Fon year 2012, and US-Ha1 year 2010) (see

Figure S1). The model underestimates GPP during these periods based on a lower PAR input, however, the observations show greater GPP despite lower input PAR. This is in line with findings of Knohl and Baldocchi (2008), where they investigated the effect of diffuse light on the forest ecosystem and discussed how diffuse radiation can lead to an increase in carbon uptake. Enhancement of GPP due to diffuse light is related to more evenly distribution and more efficient light penetration within canopy (Yuan et al., 2014). Integration of such effect in the current model by introducing a time-varying LUE (condition-

varying) (Wei et al., 2017) instead of the fixed LUE would improve the simulation. In particular, under unprecedented global warming and climate change, future changes in cloud cover and aerosol concentration are expected to modify the solar radiation and the subsequent ecosystem productivity (Durand et al., 2021; Meyer et al., 2014). Regarding LAI simulation, one limitation is that, at some points, the model cannot increase in LAI in the initial onset of LAI as fast as that of observation in the early growing period. In previous studies, it has been shown that the inclusion of non-structural carbon storage at the beginning

of green-up might help to overcome this issue (Asaadi et al., 2018) and refine the model simulation results further. Aside from the current model limitations subjected to further improvement, the model exhibits a reasonable validity and performs equally well in estimating the temporal dynamics of GPP and LAI development compared to more complex land surface and biogeochemical models. The PCM being parsimonious makes it suitable for further reaching applications in coupled models. Dynamic development of LAI is relevant to GPP estimation and beneficial for hydrologic models providing them with

prognostically driven LAI time series based on vegetation responses to temperature, particularly in the context of water budget responses to climate variability.

We aim, as a next step, to implement the presented model into the existing open-source mesoscale Hydrologic Model (mHM; Samaniego et al., 2010; Kumar et al., 2013, available at https://www.ufz.de/mhm) with a proven predictive power in simulating root-zone soil moisture dynamics (Boeing et al., 2021). The spatially simulated soil moisture derived from mHM

would provide an alternative to (limited) soil moisture observations required for GPP simulation. In particular, in the face of ongoing and future climate changes and increasing occurrence of droughts (Harper et al., 2021), reliable simulations of soil moisture are invaluable information to capture plant drought responses for the carbon cycle and climate feedbacks (Harper





et al., 2021). Finally, in doing so, we expect an improved capability of the hydrological model to represent the coupled water and carbon (i.e., GPP/LAI in this study) components.

## 4 Conclusion

In view of ongoing natural and anthropogenic changes, assessing the extent to which terrestrial plants can sequester atmospheric carbon and affect water balance through LAI implication are essential for effective climate-adaptation and resilience plans. In this study, we developed a parsimonious canopy model (PCM) with a medium level of complexity to simulate canopy GPP and LAI. In the PCM model the carbon uptake drives leaf biomass accumulation based on a mass balance approach. The model employs the light use efficiency principle in which the core concept is the conversion of absorbed photosynthetically active radiation into biomass. An integrated phenology submodel, from which allocation of carbon to and decay from the leaf pool is guided, is incorporated to predict the timing of leaf development and characterising different phenological stages. The PCM model performed reasonably well in reproducing the daily development of GPP and LAI in deciduous broad-leaved forest biome across Europe and North America. The model runs with a few required inputs; air temperature, VPD, PAR, and soil moisture (optional, recommended in dry regions and drought events). Although the proposed model runs with a number of parameters for representing the relevant processes (29 parameters without the soil moisture-related parameters), a global sensitivity analysis showed that only 10 parameters (on average across sites) were sensitive and had to be inferred via calibration. The result reaffirms that light use efficiency and specific leaf area index parameters are by far the most informative parameters in GPP and LAI simulations, respectively. The on-site calibrated maximum LUE parameter showed relatively large variation within the sites with greater maximum LUE at Dk-Sor and lower value at IT-Ro1. It implies that applying a fixed biome-specific maximum LUE does not hold applicable over different locations. Moreover, modelled GPP during growing season was shown to be almost insensitive to LAI changes. This indicates that GPP is saturated at a particular value of LAI and any further increase in LAI does not necessarily increase the resultant GPP. We also tested the robustness and generality of the underlying model structure, identifying a compromise set of model parameters applicable to all sites (region-wide). The results show that the model application is possible without site-specific calibration and yet yielding reasonable model quality. The model's skill in capturing the LAI dynamics – that was not used in the parameter inference process – further confirms the robustness of the presented model structure. Given the scarce soil moisture information, we expect that simulated soil moisture derived from a hydrological model would improve the representation of GPP simulations, particularly at semiarid regions or in drought events. We envision that the medium complexity of the presented model allows a seamless integration into large scale hydrological models to better represent ecohydrological aspects of ecosystems. We plan to implement the PCM model into the existing hydrologic models (e.g., open-source mesoscale Hydrologic Model; mHM), thereby enabling an improved representation of coupled water and carbon fluxes, in the face of a changing environment. To allow for a seamless estimation of carbon and water fluxes, we plan to include implementation of a robust regional parameter inference approach (e.g., establishing regionalized LUE parameter through a multiscale parameterization approach (Samaniego et al., 2010)) and performing extensive cross-validation experiments to ensure credible model simulations across a wide range of spatial domains.



*Code availability.*   The PCM is archived at https://doi.org/10.5281/zenodo.6373776 (Bahrami et al., 2022) (last access: 21 march 2022). It is also publicly available at https://git.ufz.de/bahrami/pcm (last access: 21 march 2022).

*Data availability.*   The flux tower dataset for DK-Sor, CA-Oas, DE-Hai, FR-Fon, IT-Ro1, US-Ha1, US-Oho, and US-MMS can be can be accessed from the FLUXNET 2015 Tier 1 at https://fluxnet.org/data/fluxnet2015-dataset/ (accessed on 20 july 2021). Data from DE-HoH
700  are available by contact: corinna.bebmann@ufz.de.

LAI field measurements for US-Ha1 can be downloaded from https://harvardforest1.fas.harvard.edu/exist/apps/datasets/showData.html?id=hf069 (accessed on 20 January 2022)

*Author contributions.*   BB coded and scripted the model. BB performed the sensitivity analysis. BB also prepared the manuscript. RK, AH, and ST were involved in interpretation of the results and discussions. All authors contributed to results discussion, reviewing, and editing the
705  manuscript.

*Competing interests.*   The contact author has declared that neither they nor their co-authors have any competing interests.

*Acknowledgements.*   We acknowledge the FLUXNET data network for providing the data at participant flux sites: DK-Sor, CA-Oas, DE-Hai, FR-Fon, US-Ha1, US-Oho, US-MMS, IT-Ro1 used in our study. We would like to thank F. Paul for providing data at DE-HoH flux site. We thank A. Klosterhalfen for helpful information on data at DE-Hai. We would also like to thank C. Rebmann (DE-HoH), M. Mund (DE-Hai),
710  T. Whitby (US-Ha1), M.P. Voyles (US-MMS) for sharing the LAI data in this study.





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

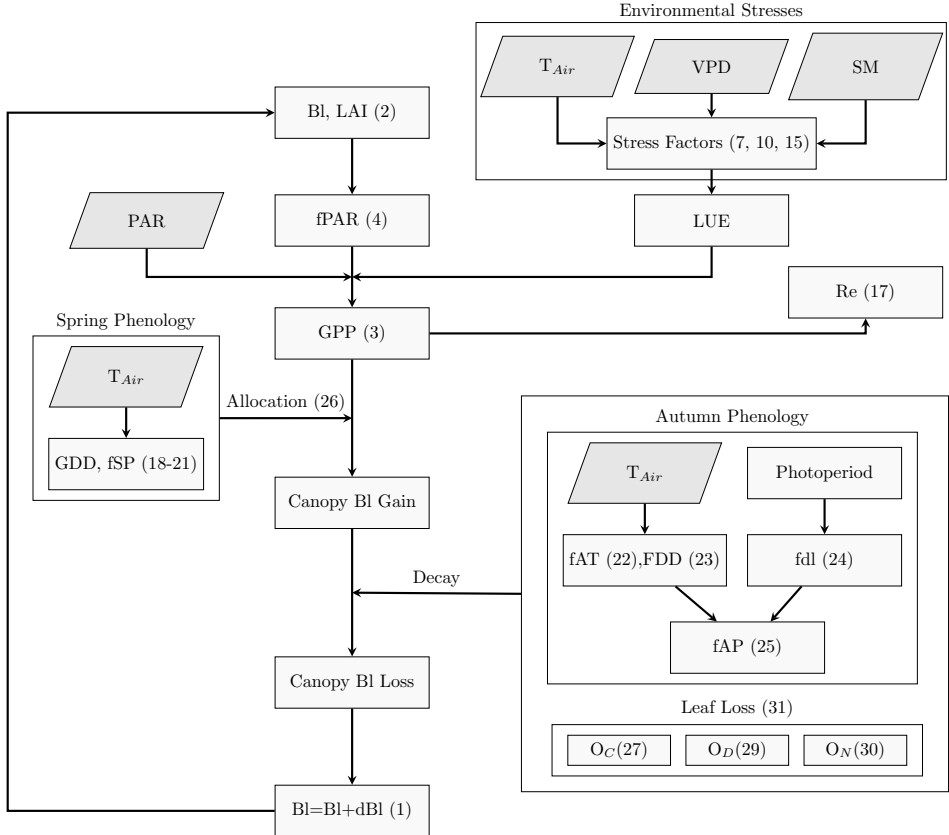

**Figure 1.** Schematic representation of the PCM model. The parallelograms indicate the model inputs; $T_{Air}$: air temperature, VPD: vapor pressure deficit, SM: soil moisture, and PAR: photosynthetic active radiation. Rectangles are the processes in the model (LUE is model parameter. Photoperiod is calculated day-length based on latitudinal distribution). Numbers refer to the corresponding equations in the text.

1050



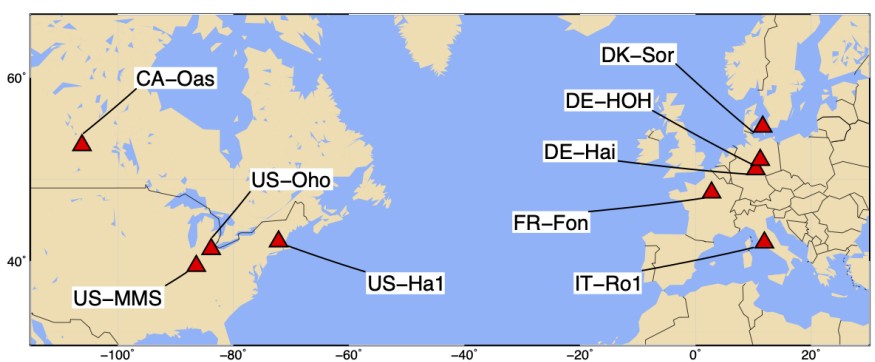

**Figure 2.** Location of the FLUXNET2015 sites investigated in this study.

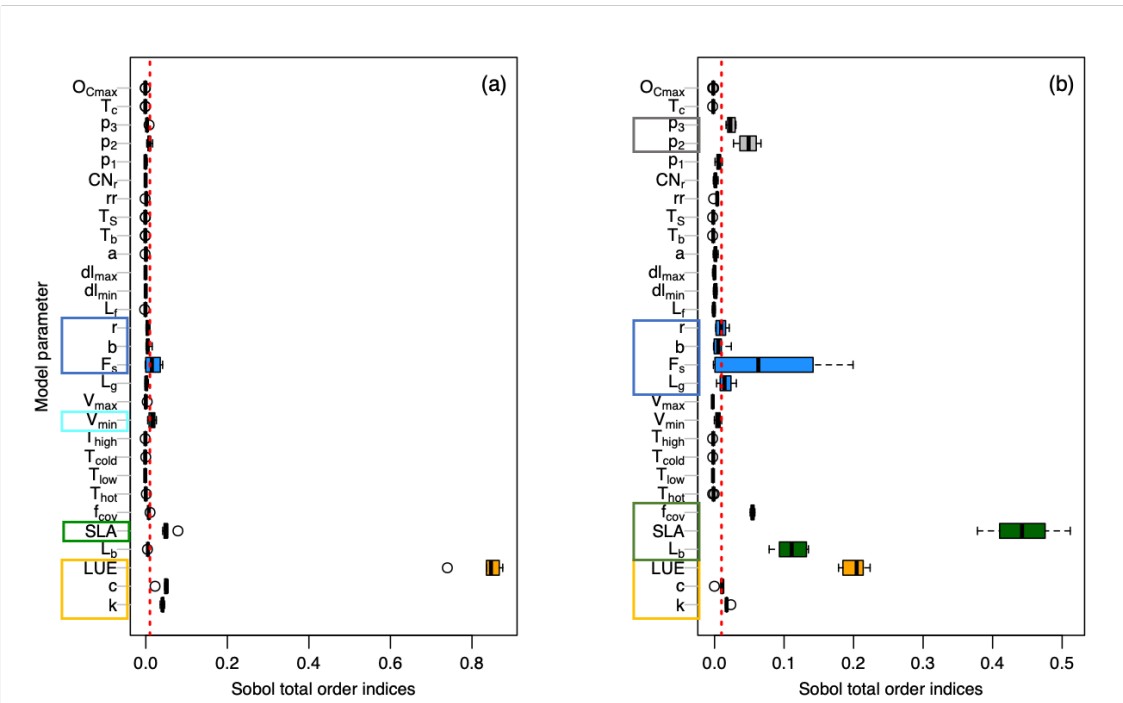

**Figure 3.** Distribution of total-order Sobol' indices for GPP (a) and LAI (b) outputs across all sites. Each colored box on the Y-axis represents parameters involved in a specific process as following: brown: GPP-related parameters (Eq. 3, 4); dark green: LAI-related parameters (Eq. 2, 26); Cyan: Environmental stresses-related parameters (Eq. 10); blue: phenology-related parameters (Eq. 18, 20, 22); grey: canopy respiration-related parameters (Eq. 17). The vertical dotted red line corresponds to the threshold of 1%





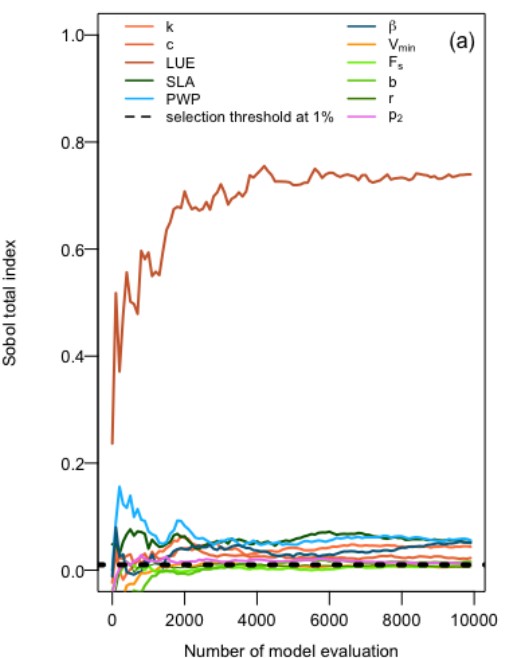
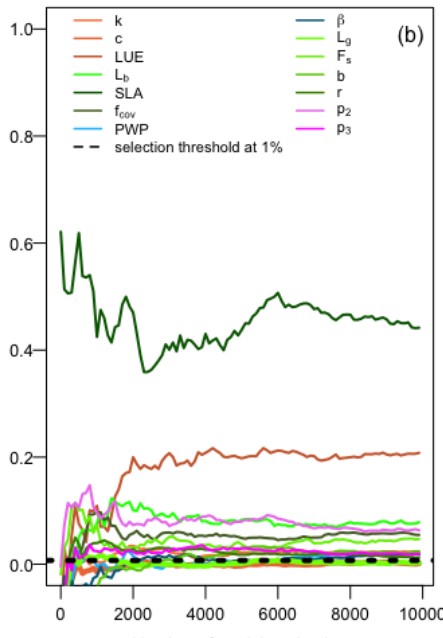

**Figure 4.** Illustration of evolution of total-order Sobol' indices (total-order indices convergence) for sensitive parameters with increasing number of samples for GPP (a) and LAI (b) outputs, at DE-HoH site taken as an example including soil moisture stress-related parameters.

**Table 1.** Descriptions of flux tower sites from FLUXNET2015 global database collection.

| Site ID | Site Name | Latitude | Longitude | Elevation(m) | Mean Annual Temperature (°C) | Mean Annual Precipitation (mm) | Time Period | Source |
|---------|-----------|----------|-----------|--------------|------------------------------|-------------------------------|-------------|--------|
| DK-Sor | Soroe | 55.48 | 11.64 | 40 | 8.2 | 660 | 1996-2014 | DOI: 10.18140/FLX/1440155 |
| CA-Oas | Saskatchewan - Western Boreal | 53.62 | -106.19 | 530 | 0.34 | 428.53 | 1996-2010 | DOI: 10.18140/FLX/1440043 |
| DE-HoH | Hohes Holz | 52.08 | 11.21 | 193 | 9.1 | 563 | 2014-2019 | Own dataset |
| DE-Hai | Hainich | 51.07 | 10.45 | 430 | 8.3 | 720 | 2000-2018 | DOI: 10.18140/FLX/1440148 |
| FR-Fon | Fontainebleau-Barbeau | 48.47 | 2.78 | 103 | 10.2 | 720 | 2005-2014 | DOI: 10.18140/FLX/1440161 |
| IT-Ro1 | Roccarespampani 1 | 42.40 | 11.93 | 235 | 15.15 | 876.2 | 2000-2008 | DOI: 10.18140/FLX/1440174 |
| US-Ha1 | Harvard Forest EMS Tower | 42.53 | -72.17 | 340 | 6.62 | 1071 | 1991-2012 | DOI: 10.18140/FLX/1440071 |
| US-Oho | Oak Openings | 41.55 | -83.84 | 230 | 10.1 | 849 | 2004-2013 | DOI: 10.18140/FLX/1440088 |
| US-MMS | Morgan Monroe State Forest | 39.32 | -86.41 | 275 | 10.58 | 1032 | 1999-2014 | DOI: 10.18140/FLX/1440083 |





**Figure 5.** Time series of observed and simulated GPP at each study site during the three last years of calibration and the two first years of verification periods. The vertical dash line marked the calibration-verification periods. The black dots indicate the tower estimated GPP. The light grey shed corresponds to the ensemble sets of modeled GPP outputs at each time step. The dark grey line refers to the median of model ensembles.



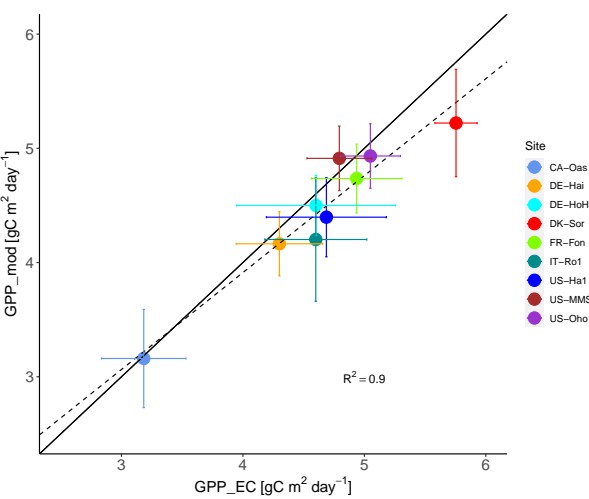

**Figure 6.** Estimated GPP based on flux tower measurements vs. modelled GPP $\pm$ standard deviation (error bars) across the 9 studied sites. The solid line indicates the 1:1 line, and the dashed line indicates the regression line. Each dot represents one of the sites and refers to site-averaged GPP over the entire available time series.

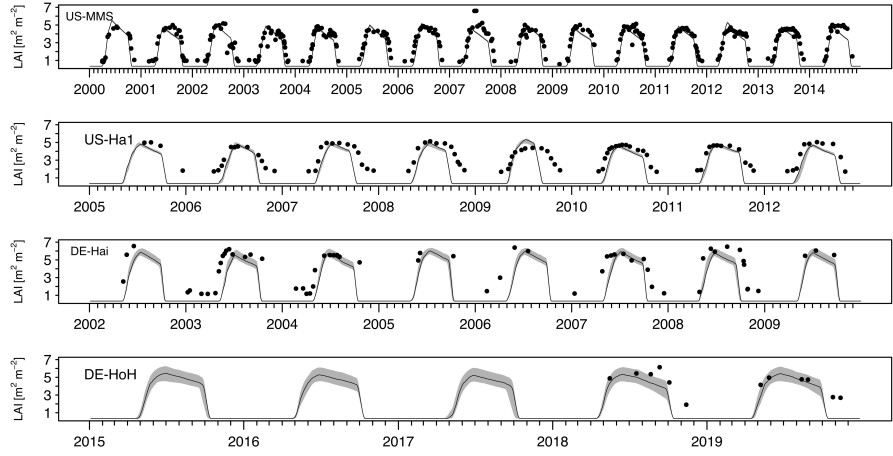

**Figure 7.** Time series of observed and simulated LAI at study flux tower sites during the common years of field measurements and simulations. The black dots indicate the field measurement LAI. The light grey shed corresponds to the ensemble sets of modeled LAI outputs at each time step. The dark grey line refers to the median of model ensembles.





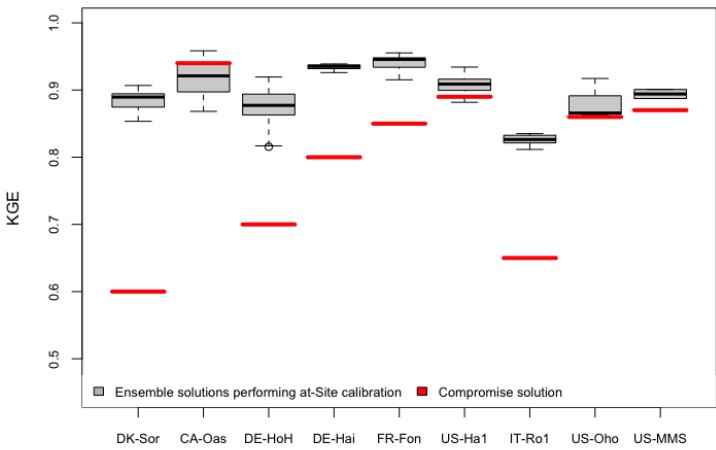

**Figure 8.** Comparison between KGE obtained from ensemble simulated GPP performing at-site calibration and the KGE obtained from compromised solution.

**Table 2.** List of input and state variables (at daily time step) in PCM.

| Input variables | Unit | Description |
|---|---|---|
| T | $^{\circ}C$ | mean air temperature |
| PPFD | $\mu mol\ m^{-2}\ s^{-1}$ | photosynthetically active radiation |
| VPD | $hPa$ | vapour pressure deficit |
| SM | % | soil moisture |
| Soil textural properties | % | sand, clay, and bulk density |
| Lat | degree | Latitude of site |
| State variables | Unit | Description |
| Bl | $gC\ m^{-2}$ | biomass of leaf |
| D | $gC\ m^{-2}$ | leaf biomass decay |
| LAI | $m^2\ m^{-2}$ | leaf area index |
| fPAR | % | fraction of photosynthetically active radiation |





**Table 3.** Model parameters in PCM

| Calibration model parameters, based on sensitivity analysis | Unit | Description | Lower Boundary | Upper Boundary | References |
|---|---|---|---|---|---|
| K | - | extinction coefficient | 0.45 | 0.60 | Ruimy et al. (1999); Yuan et al. (2007) |
| C | - | Beer–Lambert law parameter | 0.85 | 1 | Monsi and Saeki (1953) |
| LUE | $gC\,MJ^{-1}$ | light use efficiency | 1.04 | 2.25 | Cheng et al. (2014); Yuan et al. (2010) |
| $L_b$ | $m^2\,m^{-2}$ | maiximum balanced LAI | 4 | 6.5 | Fleischer et al. (2013) |
| SLA | $m^2 g^{-1}$ | specific leaf area | 0.01 | 0.03 | Kattge et al. (2011); Gim et al. (2017); Dyderski et al. (2020) |
| $f_{cov}$ | % | vegatation fractional coverage per unit area | 0.60 | 0.95 | Fluxnet site description |
| PWP | % | permanent wilting point | 7 | 13 | Intermediate output of PCM model |
| $\beta$ | - | root distribution coefficient | 0.966 | 1 | Jackson et al. (1996) |
| $v_{min}$ | $hPa$ | mean VPD at which LUE = $LUE_{potential}$ | 6.5 | 10 | Heinsch et al. (2003); Cheng et al. (2014) |
| $L_g$ | $DegreeDay$ | phenological growing length | 300 | 450 | Yue and Unger (2015) |
| $F_s$ | $DegreeDay$ | phenological threshold for leaf fall | -500 | -112 | Yue and Unger (2015), calibrated |
| b | $DegreeDay$ | phenological parameter for budburst threshold $G_b$ | 440 | 660 | Yue and Unger (2015) |
| r | - | phenological parameter for budburst threshold $G_b$ | -0.012 | -0.008 | Yue and Unger (2015) |
| $p_2$ | - | 2nd parameterin Arrhenious equation | 44.96 | 67.44 | Sitch et al. (2003) |
| $p_3$ | - | 3rd parameterin Arrhenious equation | 36.96 | 55.44 | Sitch et al. (2003) |
| Fixed model parameters based on sensitivity analysis | | | | | |
| FC | % | field capacity | 23 | 23 | Intermediate output of PCM model |
| scw | - | critical threshold value of soil moisture | 0.4 | 0.4 | Granier et al. (1999) |
| $T_{hot}$ | $^{\circ}C$ | mean air temperature of warmest month | 19 | 19 | Rödig et al. (2017); Sitch et al. (2003) |
| $T_{low}$ | $^{\circ}C$ | low temperature limit for $CO_2$ assimilation | -2 | -2 | Rödig et al. (2017); Sitch et al. (2003) |
| $T_{cold}$ | $^{\circ}C$ | mean air temperature of coldest month | 10 | 10 | Rödig et al. (2017); Sitch et al. (2003) |
| $T_{high}$ | $^{\circ}C$ | high temperature limit for $CO_2$ assimilation | 38 | 38 | Rödig et al. (2017); Sitch et al. (2003) |
| $v_{max}$ | $hPa$ | mean VPD at which LUE = 0 | 25 | 25 | Heinsch et al. (2003); Cheng et al. (2014) |
| $L_f$ | $DegreeDay$ | phenological falling length | 410 | 410 | Yue and Unger (2015) |
| $dl_{min}$ | $minutes$ | phenological day length threshold for leaf fall | 585 | 585 | Yue and Unger (2015) |
| $dl_{max}$ | $minutes$ | phenological day length threshold for leaf fall | 695 | 695 | Yue and Unger (2015) |
| a | $DegreeDay$ | phenological parameter for budburst threshold $G_b$ | -110 | -110 | Yue and Unger (2015) |
| r | - | phenological parameter for budburst threshold $G_b$ | -0.01 | -0.01 | Yue and Unger (2015) |
| Tb | $^{\circ}C$ | base temprature for budburst occurrence | 5 | 5 | Yue and Unger (2015) |
| Ts | $^{\circ}C$ | base temprature for senescence occurrence | 20 | 20 | Yue and Unger (2015) |
| CNr | $gC\,gN^{-1}$ | leaf C:N ratio | 25 | 25 | White et al. (2000) |
| $p_1$ | - | 1st Arrhenious parameter | 308.56 | 308.56 | Sitch et al. (2003) |
| $T_c$ | $^{\circ}C$ | temperature threshold for determining cold stress | 5 | 5 | Melton and Arora (2016) |
| rr | $gC\,gN^{-1}$ | leaf respiration coefficient | 0.066 | 0.066 | Kattge et al. (2011); Sitch et al. (2003); Rödig et al. (2017) |
| $O_{Dmax}$ | $day^{-1}$ | maximum drought stress loss rate | 0.15 | 0.15 | Melton and Arora (2016) |
| $O_{Cmax}$ | $day^{-1}$ | maximum cold stress loss rate | 0.005 | 0.005 | Melton and Arora (2016) |



**Table 4.** Summary statistics for the comparison between model estimated GPP and tower estimated GPP at different sites. Statistics include KGE, root mean square error (RMSE), and $R^2$. GPP units are [$g\ C\ m^{-2}\ d^{-1}$]. The statistics refer to ensemble medians of model estimated GPP.

| Site | Calibration | | | | Verification | | | |
|---|---|---|---|---|---|---|---|---|
| | Period | KGE | RMSE | $R^2$ | Period | KGE | RMSE | $R^2$ |
| DK-Sor | 2007-2010 | 0.89 | 2.09 | 0.89 | 2011-2013 | 0.89 | 2.15 | 0.89 |
| CA-Oas | 1997-2004 | 0.92 | 1.5 | 0.89 | 2005-2010 | 0.90 | 1.4 | 0.91 |
| DE-HoH | 2015-2017 | 0.88 | 1.8 | 0.88 | 2018-2019 | 0.75 | 2.5 | 0.80 |
| DE-Hai | 2001-2015 | 0.93 | 1.9 | 0.85 | 2016-2018 | 0.91 | 2.01 | 0.84 |
| FR-Fon | 2006-2010 | 0.95 | 1.7 | 0.91 | 2011-2014 | 0.91 | 1.94. | 0.85 |
| US-Ha1 | 2004-2008 | 0.92 | 2.03 | 0.86 | 2009-2012 | 0.88 | 2.56 | 0.80 |
| IT-Ro1 | 2002-2004 | 0.79 | 2.45 | 0.65 | 2005-2006 | 0.86 | 1.87 | 0.78 |
| US-Oho | 2005-2010 | 0.87 | 2.22 | 0.85 | 2011-2013 | 0.85 | 2.39 | 0.82 |
| US-MMS | 2000-2007 | 0.9 | 2.1 | 0.85 | 2008-2014 | 0.89 | 1.9 | 0.87 |

**Table 5.** Summary statistics for the comparison between model estimated LAI and Field measurement LAI at different sites. Statistics include $R^2$ and RMSE. LAI units are [$m^{-2}m^{-2}$]. The statistics refer to ensemble medians of model estimated LAI.

| Site | Period | RMSE | $R^2$ |
|---|---|---|---|
| US-MMS | 2000-2014 | 0.96 | 0.90 |
| US-Ha1 | 2005-2012 | 1.58 | 0.85 |
| DE-Hai | 2002-2009 | 2.21 | 0.78 |
| DE-HoH | 2018-2019 | 1.4 | 0.90 |