# Peer review of "Developing a Parsimonious Canopy Model (PCM v1.0) to Predict Forest Gross Primary Productivity and Leaf Area Index on deciduous broad-leaved forest"

_Geoscientific Model Development, 2022_

## Author Comment (AC1)

Dear Editor,

We wish to thank you and the referees for your precious time in reviewing our paper and providing valuable comments. It was your valuable and insightful comments that led to possible improvements in the current version. Following the editorial office suggestions, we combined the Supplementary figure's captions and the figures themselves into a single pdf supplement file, we also checked and revised the figures to ensure that figures are accessible to colorblind readers.
We have carefully considered the comments and tried our best to address every one of them. Below we provide the point-by-point responses to referees' comments. Texts in italic are the referees' comments (C), those in black bold style are our responses (AR), and texts marked in red are relevant changes in the manuscript. A marked-up version, showing the changes in the revised manuscripts, has also been prepared to be submitted accordingly. The page and line numbers in this letter refer to the marked-up version. We hope that you will find the changes satisfactory.

Sincerely,

Bahar Bahrami on behalf of the co-authors
bahreh.bahrami@ufz.de
Department of Computational Hydrosystems
UFZ, Leipzig, Germany

**Referee # 1**

Dear Referee,

Thank you for your time and attentions on this work. The comments and suggestions are very useful to improve our manuscript. We paid detailed attention to all comments and have addressed all of them below accordingly. Please find our response in the supplement.

**General comments**

*Bahrami et al. developed a forest development model requiring few parameters linked with a phenology submodel predicting gross primary productivity (GPP) and leaf area index (LAI). They evaluated model performance at a selection of FLUXNET sites and performed a parameter sensitivity analysis determining the most sensitive parameters and optimal site-specific values as well as a set of compromise parameter values for larger regions.*

*The model can be coupled with a hydrologic model, which could improve both water and carbon flux simulations. The model is well developed and performs reasonably well at broad-leaved forest sites. The language and description of certain parts of the manuscript, however, should be improved before publication. Especially the Introduction and parts of the Results and Discussion are unclear and should be better explained. I would suggest rewriting most of the Introduction section to explain the cited literature and its relevance to the manuscript better. Generally, the model is explained well, but certain parts of the description can be made clearer (see specific comments below).The manuscript contains a lot of technical corrections (typos and grammatical mistakes), which should be corrected.*

**We appreciate the reviewer's suggestions! We have revised the manuscript based on all the general and specific comments raised by the reviewer and rewrote most of the introduction. We have also carefully proofread the manuscript and have made several technical corrections, and hope that the revised manuscript will meet with your requirements.**

**Specific comments and technical corrections**

*C 1) l. 5: Add "the" water cycle*
**AR 1) We added it to the sentence [Pg. 1, l. 5].**
"…which is of critical significance in and closely linked to the water cycle."

*C 2) l. 38: Use "sequestering" instead of "sequestrating" (also in l. 69, 79).*
**AR 2) We have modified the sentence as follows [Pg. 3, l. 76], [Pg. 4, l. 95], [Pg. 4, l. 108].**
"… the extent to which ecosystems are capable of sequestering it, …"
"… the rubisco enzyme uses the ATP energy from the light response to sequester the atmospheric carbon dioxide …"
"The amount of sequester carbon as biomass …"

*C 3) l. 43: In "60% of the global net forest sink" do you mean the "global net carbon sink"?*
**AR 3) Thank you for this remark. We meant the temperate broad-leaved forest contribute around 60% of the global net carbon sink among all type of forests. We have made the sentence clearer now as below [Pg. 2, l. 50].**
"Generally, forests are recognized as biomes with high carbon sequestration capacity (Lal and Lorenz, 2012) where temperate broad-leaved forest contribute to approximately 60% of the global net carbon sink of forests (Pan et al., 2011; Reinmann and Hutyra, 2017)."

*C 4) l. 44: Specify what kind of ecosystems you mean; I haven't heard it defined as "vegetation GPP", just GPP. You could say "vegetated ecosystems". "Plant photosynthesis" already implies that it comes from vegetation.*

**AR 4:) Thank you for the suggestion. We have accordingly revised the sentence [Pg. 3, l. 78].**

"The total carbon uptake from the atmosphere into the vegetated ecosystems by plant photosynthesis is known as gross primary production (GPP)."

*C 5) l. 45: GPP and ecosystem respiration are of similar magnitude and which one is larger depends on whether the ecosystem is a sink or source of $CO_2$. I would rephrase this.*

**AR 5) Thanks for this suggestion. We modified the sentence to make it clearer [Pg. 3, l. 79].**

"GPP is the primary driver of the land carbon sink (Spielmann et al., 2019; Zhou et al., 2021) and the largest flux within the carbon cycle (Schaefer et al., 2012; Foley and Ramankutty, 2003)."

*C 6) l. 46: Be more specific what you mean with "has a direct effect on moderating climate and environment", especially the effect on the environment.*

**AR 6) Thanks for the suggestion. We have added sentences to better explain the effect on climate and environment [Pg. 3, l. 81].**

"Accurate estimation of GPP directly influences carbon budget assessments as well as estimates of the amount of stored carbon in the plant leaf pool. Accurate carbon budget assessment, in turn, promotes understanding of the feedbacks between the terrestrial biosphere and the climate system (Zhou et al., 2021; Huang et al., 2022)."

*C 7) ll. 46-50: Be more specific about what "adverse effects of a changing climate" you mean! The second part of the sentence applies to any climatic conditions not only under a changing climate. To relate this part of the sentence to the first part about climate change, discuss its effects on temperature, water availability, radiation, etc. Otherwise, the reason for mentioning a changing climate here is unclear.*

**AR 7) Thank you for the comment. We have revised the paragraph following your valuable suggestion [Pg. 2, l. 42].**

"Vulnerability due to climate change can be attributed to different ecosystem stresses (Nathalie et al., 2006; Cholet et al., 2022) including high temperatures that decrease enzymes activity and the rate of carbon uptake as well as soil water limitation causing hydraulic failure or carbon starvation, reducing plant photosynthetic capacity, and early senescence (Imadi et al., 2016) in temperate forest ecosystems. In addition to these stresses, some environmental changes such as radiation change associated with increased cloudiness or atmospheric aerosols can also increase plant productivity, e.g. due to an increased fraction of diffused radiation (Knohl and Baldocchi, 2008)".

*C 8) l. 50: Favourable climate in what respect? I'm not sure what you want to express here and how, for example, the winter season is favourable for the vegetation.*

**AR 8) Thank you for this comment. We had put "favorable" mistakenly instead of "temperate". It has been revised as follows [Pg. 3, l. 59].**

"Temperate DBF biomes are characterized by having a temperate climate with four distinct seasons and a temperature-driven canopy structure."

*C 9) l. 51: "The plant canopy capacity and seasonality are expressed by leaf area index (LAI)" -¿ Rephrase this! What exactly do you mean with "plant canopy capacity" and LAI itself does not express seasonality. Changes in LAI do.*

**AR 9) Thank you for this comment. Here by plant canopy capacity we meant the capacity to exchange the fluxes. We have rephrased the sentence accordingly [Pg. 3, l. 61].**

"The plant canopy capacity for water and carbon exchange is strongly related to seasonal variation in leaf development (Seo and Kim, 2021)."

*C 10) l. 52: Reference? Maybe make it clearer that with "total green leaf area" you mean two-sided, as opposed to one-sided leaf area in broadleaf canopies, or total needle surface area in conifers.*

**AR 10) We revised the sentence and added the reference as follows [Pg. 3, l. 62].**

"Leaf area index (LAI) is a dimensionless quantity, defined as one-sided area of green leaf per unit horizontal ground surface area (Nathalie, 2003; Fang et al., 2019)."

*C 11) l. 53: Be more specific or add a reference here.*

**AR 11) We made the sentence more specific and added a reference [Pg. 3, l. 64].**

"LAI can be estimated either by direct field measurements, inferred using remote sensing or be simulated by vegetation carbon cycle models (Fang et al., 2019)."

*C 12) ll. 54-55: Rephrase this to make it clearer! Yes, LAI affects transpiration, but GPP does as well.*

**AR 12) Thanks for the comments. We have rephrased the sentence to make it clearer [Pg. 3, l. 67].**

"LAI is a key biophysical plant variable, representing vegetation state, affecting not only the sequestration of carbon from the atmosphere via photosynthesis but also the release of water to the atmosphere through transpiration (Fang et al., 2019)."

*C 13) ll. 56-57: If you mention water balance components affected by LAI, I would include canopy evaporation as well.*

**AR 13) Thanks for this comment. Canopy evaporation and interception are added to**

**the sentence [Pg. 5, l. 129].**

"(e.g., plant transpiration and canopy evaporation)"

*C 14) l. 68: Unnecessary to have both "later" and "in the next step". Could just say "in the dark reactions of the Calvin cycle, ...".*

**AR 14) Thanks. It has been revised accordingly [Pg. 4, l. 95].**

"In the dark reactions of the Calvin cycle, the rubisco enzyme uses the ATP energy from . . . "

*C 15) l. 71: "specifically at scales larger than the leaf level" -¿ Above, you only mention that GPP is determined at the leaf level in the EK approach. You don't say how it is up-scaled to the canopy level or larger scales.*

**AR 15) Thanks for the comment. We agree with the reviewer. The sentence with the last part was confusing and it has been removed during revision [Pg. 4, l. 96].**

"This approach requires the specification of a relatively large number of parameters for governing processes."

*C 16) ll. 76-77: Rephrase to make it clearer! It isn't clear that you mean that APAR is a product of PAR and fPAR, which is the biome-specific LUE parameter.*

**AR 16) We have revised the sentence to make it clearer. The APAR is a product of PAR and fPAR. We revised the sentence as follows [Pg. 4, l. 103].**

"In this approach, ecosystem GPP is a function of absorbed photosynthetically active radiation (APAR) and a biome specific LUE parameter (Gamon, 2015; Springer et al., 2017). APAR is a product of incident photosynthetically active radiation (PAR) and the fraction of PAR (fPAR) absorbed by plant leaves."

*C 17) l. 78: "The LUE" -¿ Say either "the LUE parameter" or "fPAR". You aren't talking about LUE itself here.*

**AR 17) Thanks for pointing this out. Indeed, this should be mentioned [Pg. 4, l. 106].**

"The LUE parameter corresponds to the vegetation conversion efficiency of solar radiation into biomass and is defined as the amount of carbon produced per unit of absorbed PAR (Monteith, 1977; Yuan et al., 2014)."

*C 18) ll. 81-82: "CFLUX (Turner et al., 2006), EC-LUE (Yuan et al., 2007), MODIS-GPP (Running et al., 2004), VPM (Xiao et al., 2004), and CASA" -¿ Define what these abbreviations stand for!*

**AR 18) We have now added the complete name of the models [Pg. 4, l. 113].**

"carbon cycle model (CFLUX). . . , eddy covariance- light use efficiency (EC-LUE) . . . , moderate resolution imaging spectroradiometer-gross primary production (MODIS-GPP) . . . , vegetation photosynthesis model (VPM) . . . , and Carnegie-Ames-Stanford Approach

(CASA)..."

*C 19) l. 86: Why specifically central Europe? If you mention it, explain why as well!*
**AR 19) Thanks for the question. Central Europe was mistakenly put here. We have rephrased the sentence as follows [Pg. 4, l. 117].**
"These two key biophysical variables are generally sensitive to cloud contamination leading to gaps in their temporal and spatial coverage throughout the year"

*C 20) ll. 86-89: Unclear what you mean. Be more specific!*
**AR 20) Thanks for the comment. We have revised the sentence as follows [Pg. 4, l. 120].**
"These gaps are sources of uncertainty in satellite-based fPAR and LAI products which, in turn, may induce errors in quantifying GPP (Rahman et al., 2022)."

*C 21) ll. 89-101: The purpose of this paragraph isn't really clear, as several different models are mentioned, but their limitations aren't clearly explained!*

**AR 21) Thanks for the comment. We have revised the paragraph to make it clearer. Here, in general, we wanted to mention previous efforts of models simulating LAI using GPP. The limitations for TETIS-VEG are that it is only applicable for evergreen forests, and also that the source code is not freely available. Regarding, the SGPD-TS model, although it simulates LAI but its limitation is that it uses a linear relationship between steady-state GPP and LAI. In this way, GPP is used as a proxy of LAI which utilizes a conversion ratio when maximum GPP has been reached. However, it has been earlier shown that maximum GPP saturates at LAI values above 4 $m^2m^{-2}$ (Lee et al., 2019); and this may potentially introduce uncertainty during simulation of LAI when the LAI of stands exceed values of 4 $m^2m^{-2}$. Many of these models have been developed and validated at specific sites and their broader applicability across a diverse range of climatic conditions has yet been not demonstrated [Pg. 5, l. 131].**
"The LUE principle and leaf growth have been successfully implemented in the TETIS-VEG ecohydrology model (Francés et al., 2007; Pasquato et al., 2015). The TETIS-VEG model is, however, adapted for evergreen forest biome. In other words, the TETIS-VEG model lacks representation of a dynamic leaf phenology relevant in the deciduous broad-leaved forests. Another approach to simulate GPP and LAI is adopted in the simplified growing production day time-stepping scheme (SGPD-TS) model (Xin et al., 2019). The SGPD-TS model, however, does not represent leaf growth and allocation to leaf pool, but establishes a linear relationship between steady-state GPP and LAI. In this way, GPP is used as a proxy of LAI, utilizing a conversion ratio when maximum GPP has been reached. However, it has been shown that simulated GPP saturates at high LAI values (e.g., above 4.5 $m^2m^{-2}$ (Lee et al., 2019) and (Pan et al., 2021)). High LAI values are often common

in deciduous broad-leaved forests, thus, relying on maximum GPP to derive LAI might introduce a bias at elevated LAI. Another more general challenging aspect for these models is the identification of model parameters that are site or location specific. Previous applications often have been limited to one calibration site (Francés et al., 2007); but they need to be thoroughly cross-validated for their applicability across a diverse range of climatic conditions"

*C 22) l. 105: Explain what specifically you mean with "readily available observational datasets across eddy flux tower stations"*

**AR 22) Thanks for this comment. We meant the most common available data set among eddy flux tower stations that are easy to obtain, i.e., can be downloaded. We made the sentence clearer by specifying the name of variables [Pg. 5, l. 150].**

"The parsimonious approach and level of model complexity are designed to make use of readily available observational dataset for abiotic forcing across eddy flux tower stations such as air temperature, vapour pressure deficit, soil moisture, photosynthetic photon flux density."

*C 23) ll. 180: "changes of vapour pressure deficit" should be "changes in vapour pressure deficit".*

**AR 23) Thanks for the comment. We changed the sentence accordingly [Pg. 8, l. 229].**

"The canopy photosynthesis rate is strongly related to changes in vapour pressure deficit …"

*C 24) l. 200: It is unclear what you mean with "using the cumulative root fraction up to each layer". What is the cumulative root fraction used for, if the root fraction for each layer is multiplied by the soil moisture content of that layer?*

**AR 24) Thanks for raising this question. For the calculation of a root-zone weighted soil moisture, we used information on depth (layer) specific soil moisture and fraction of roots in each soil layer. The latter is estimated using formulations provided by Jackson et al., 1996 – in which the cumulative root fraction at a specified depth can be expressed by an asymptotic (power law) equation along with a biome specified parameter (in our case for DBF is 0.966; see Eq. 11). We then deduce the root fraction for a specific soil layer from this cumulative root fraction estimates (see Eq. 12). Another point is that the root fractions are normalized to 100%. We have created a schematic representation here to better explain this averaging part [Pg. 9, l. 249].**

[Figure]

| Depth[cm] | $Rc_i$ | $Ri_i$ | $\theta_i$ | $\theta_t$ ($\theta_i * Ri_i$) |
|---|---|---|---|---|
| 0 | 0 | 0 | 0 | 0 |
| 0-10 | 0.29 | 0.31 | 13 | 4..1 |
| 10-20 | 0.50 | 0.22 | 14 | 3.1 |
| 20-30 | 0.64 | 0.15 | 12 | 1.9 |
| 30-40 | 0.74 | 0.11 | 12 | 1.3 |
| 40-50 | 0.82 | 0.07 | 15 | 1.2 |
| 50-80 | 0.93 | 0.012 | 20 | 2.5 |

Normalized: dividing by 0.93

Ex. For the last day of data

$$\sum \theta_i = \theta \ (daily\ effective)$$

"Then, to estimate the root fraction in each individual layer (Eq. 12), we use the calculated cumulative root fraction up to each layer subtracted from the corresponding fraction of the previous layer (see Eq. 11)."

*C 25) l. 228: "photosynthetical", not "photosynthetically".*
**AR 25) Thanks, modified to photosynthetical [Pg. 10, l. 279].**
"is the sum of photosynthetical carbon uptake by plants (GPP)"

*C 26) l. 251: Do you mean "growing season length"?*
**AR 26) Here we refer to *Lg* parameter which is a threshold in degree day for the duration of leaf growing length from budburst day up to the day of maximum canopy leaf cover. This parameter, with the same description, is adapted from Yue and Unger, 2015. We revised the sentence to make it clearer [Pg. 10, l. 302].**
"The $L_g$ parameter is a calibrated constraint in degree day, representing the period of leaf growth from budburst to maximum leaf cover (Yue and Unger, 2015)"

*C 27) l. 286: Why have "used" twice in the sentence?*
**AR 27) Thanks for pointing this typing mistake. We have deleted the second one [Pg. 12, l. 338].**
"There are two widely used allocation schemes in vegetation models based on: ... "

*C 28) l. 288: It should be "BIOME-BGC" (also in l. 369).*
**AR 28) Thanks. We modified the text [Pg. 12, l. 340], [Pg. 15, l. 434].**
"... or BIOME-BGC (Hidy et al., 2016)." "... (BIOME-BGC; Hidy et al.,2016)"

*C 29) Equation 28: It should be 0 for Tc ≤ T(t).*
**AR 29) Thank you for noticing. It was indeed a typo in the text. We modified the third part of Eq. 28 as following [Pg. 13, l. 371].**

"0, Tc $\leq$ T(t)"

*C 30) l. 339: Either use "we" or remove the "and" and add a ".".*
**AR 30) Thanks for the comment. We removed "and" [Pg. 13, l. 391].**
"This study focuses on deciduous broad-leaved forests biome type. We selected tower sites distributed over Europe and North America to ensure a representative spatial coverage"

*C 31) l. 342: Be more specific what you mean with "long missing data at some sites".*
**AR 31) Indeed this sentence was not very clear. We meant that in some of the FLUXNET sites there were continually long periods of missing data such as years of missing data for PPFD where we excluded those years. For example, in the US-Ha1 site, even though the dataset in FLUXNET web page are available from 1991 to 2012, there is a long period of missing data for PPFD from 1991 to 2003. Therefore, our simulation starts in 2003. We added the following sentences to make it clearer [Pg. 14, l. 394].**
"We further screened out the data at each site to the years with minimal gap in input data. For example, there were some long period of gaps (i.e., years) within the continuously recorded FLUXNET dataset for photosynthetic photon flux density (PPFD), which we excluded those years in the simulations (e.g., a continuous period of missing PPFD in the US-Ha1 dataset from 1991-2003)"

*C 32) l. 350-351: Make it clearer whether the soil moisture and soil texture variables are optional or required for the model.*
**AR 32) Thanks for this suggestion. The text is now revised. In fact, the soil moisture (SM) and soil texture variables are optional for running the model. We had SM related information only for the DE-HoH site and therefore the application of soil moisture module was possible for only this site. In contrast, the model was run without the SM module for the other studied sites. While revising the text, we removed the word "required". Based on the change in this sentence we also removed "However" at the beginning of the next sentence [Pg. 14, l. 408].**
"Soil moisture (SM) and soil textural properties need to be provided to the model, if the model should also consider soil moisture stress. We investigated . . ."

*C 33) l. 359: "obtained" not "collected" via personal communication.*
**AR 33) Thanks for the suggestion. We changed "collected" to "obtained" [Pg. 14, l. 418].**
"The LAI field measurements were obtained via personal communication to site contact persons: . . ."

*C 34) l. 360: Maybe say "s subset of 4 sites was selected based on data availability" instead of "based on the responses a subset of 4 sites are used".*

**AR 34) Thanks for suggestion. We modified the text [Pg. 14, l. 419].**

"; and a subset of 4 sites (DE-HoH, DE-Hai, US-MMS, and US-Ha1) was selected based on data availability"

*C 35) l. 364: What do you mean with "closest methods"? Are these methods both used at the same site or is one of the methods used at each site?*

**AR 35) We meant at one of the sites the fisheye method is used and at the others the LAI-2000 method. According to Ariza-Carricondo et al. (2019), these two methods agree very well with each other providing nearly similar of LAI across different sites. We have revised the texts accordingly [Pg. 14, l. 421].**

"The observation-based LAI data were obtained using common procedures with either the LAI-2000 instrument (Gower and Norman, 1991) at the DE-Hai, US-MMS, and US-Ha1 or the fisheye (DHP) technique ((Bonhomme, R. and Chartier, P., 1972; Ariza-Carricondo et al., 2019)) at the DE-HoH site, respectively. These two methods agree very well according to Ariza-Carricondo et al. (2019) and are thus considered to yield comparable values also across different sites."

*C 36) ll. 368-371: Explain the different water stress functions better instead of just mentioning their names.*

**AR 36) Thanks for the suggestion. We also noticed that we had put the CASA model in a wrong category. We also added FORMIND in the text where soil moisture stress is considered in the model. We revised the text and add to the sentence accordingly to explain this part better [Pg. 15, l. 429].**

"The impact of water availability on the canopy photosynthesis (i.e., soil water deficit and atmospheric water deficit), in vegetation models is structured in two ways: individually or in combination with each other. Recently, plant hydraulic theory has also been introduced to reflrct the vegetation water stress in Community Land Model (CLM5), which is beyond the scope of this study (Kennedy et al., 2019). In some models, water stress is quantified as an overall stress from both atmosphere and soil ((GLO-PEM; Prince and Goward, 1995), (BIOME-BGC; Hidy et al., 2022)). For instance, in the GLO-PEM model the water stress condition is reflected by an estimated and potential evapotranspiration, a relative drying rate scalar for potential water extraction, and a volumetric soil moisture content (more details together with equations can be found in (Zhang et al., 2015)). Some other models account for the water stress only due to the atmospheric drought ((CASA; Potter et al., 1993), (MOD17 algorithm; Running et al., 2000)). For example, in the MOD17 algorithm, only the atmospheric variable VPD and its two parameters, $v_{min}$ and $v_{max}$, are used to calculate water stress factor to predict GPP (Running et al., 2000). In some other models such as FORMIND (Fischer et al., 2016) and EC-LUE (Yuan et al., 2007) only the soil moisture deficit is reflected. For instance, in the FORMIND model, the impact of atmospheric water deficit (VPD impact) is not presented; but the soil moisture deficit is represented by volumetric soil water content and soil parameters (soil field capacity, permanent wilting point, and minimum soil water content).."

*C 37) l. 375: It should be "in 2018".*
**AR 37) Modified [Pg. 15, l. 447].**
"... on simulated GPP over the DE-HoH site during the drought in 2018."

*C 38) l. 390: In "the literature".*
**AR 38) Modified [Pg. 16, l. 462].**
"In this study during the GSA, the parameters vary over boundaries reported in the literature's."

*C 39) l. 420: Do you not spin up the model for a longer time period? How can soil C and other C pools be spun up after one year or do you fully spin up the model with the default parameter values only?*
**AR 39) Thanks for the question. Here we focus on temperate forests and only on the above ground carbon pool confined to the canopy and leaf pool. The leaf pool at the end of each annual active growing season reaches to zero and the next year start almost from a bare canopy and zero carbon in the leaf pool. Therefore, we do not spin up the model for long period where it is indeed more relevant for compartment such as soil carbon, which we are not simulating the carbon in soil or other pools.**

*C 40) l. 466: "in (Eq. 3)" -¿ either remove the brackets or put the "in" into the brackets as well.*
**AR 40) Modified [Pg. 18, l. 542].**
"LUE in (Eq. 3)."

*C 41) l. 468: Add "the" in front of "Farquhar photosynthesis scheme".*
**AR 41) Added [Pg. 18, l. 545].**

*C 42) l. 469: Add "the" in front of "photosynthesis process".*
**AR 42) Added [Pg. 18, l. 546].**

*C 43) l. 474: Switch order to "also showed".*
**AR 43) Added [Pg. 18, l. 550].**
The multiplicative coefficient of canopy reflectance, C, and the light extinction coefficient, k, parameters in the fPAR formulation (Eq. 4) based on Lambert-Beer's law also showed substantial sensitivities.

*C 44) ll. 474-475: Rephrase! I don't think you need both "typically" and "by default".*

*Make it clearer what you're doing differently, if you mention that the parameters are "typically" fixed.*

**AR 44) "Typically" is removed. We added to text to make it clearer. This investigation helps to explore the model sensitivities to the often hidden parameters with the possibility to properly constrain the model [Pg. 18, l. 551].**

"Notably, these parameters are fixed to constant values by default in the fPAR formulation in similar studies (e.g., Xiao et al. (2004) and Xin et al. (2019)); whereas, here, we let these parameters ($C$ and $k$) vary at $\pm 20\%$ level of their fixed values."

*C 45) l. 479: Instead of saying "the impact", specify what kind of impact (e.g., strong, weak) and say "VPD" or "the VPD variable".*

**AR 45) Revised [Pg. 18, l. 556].**

"the strong impact of VPD ..."

*C 46) l. 480: It should be "the" next environmental factor constraining "GPP".*

**AR 46) Modified [Pg. 18, l. 557].**

"The next environmental factor constraining GPP ..."

*C 47) l. 481: It should be "at the DE-HoH site".*

**AR 47) Modified [Pg. 18, l. 558].**

*C 48) l. 484: Remove "the" in front of $\theta_r$ (also in l. 486).*

**AR 48) Modified [Pg. 19, l. 562].**

*C 49) l. 485: Add "a" in front of "soil matric potential".*

**AR 49) Added [Pg. 19, l. 562].**

*C 50) ll. 487-489: Be more specific what you would use as parameters?*

**AR 50) Thank you for this remark. With the text mentioned there, we wish to emphasise that empirical coefficients in pedo-transfer functions, linking soil textural properties (like sand or clay contents) with hydraulic characteristics (like permanent wilting points, field capacity), can be considered as parameters. To this end, we have provided references of previous studies also emphasising this aspect. We have revised the text as follows [Pg. 19, l. 565].**

"Pedo-transfer functions (PTFs) link soil textural properties (e.g., sand, clay contents) to soil parameters (e.g., $\theta_r$) and various functional forms have been developed in past decades (Van Looy et al., 2017). Empirical coefficients of PTFs can also be regarded as model parameters (Samaniego et al., 2010; Kumar et al., 2013; Schweppe et al., 2021)."

*C 51) ll. 490: Remove "the" in front of "simulated GPP".*

**AR 51) Modified [Pg. 19, l. 571].**
".. is also a major contributor to simulated GPP ..."

*C 52) l. 491: Add "the" in front of "LAI calculation".*
**AR 52) Modified [Pg. 19, l. 572].**
".. the LAI calculation ..."

*C 53) l. 497: Say "at some sites", not "in".*
**AR 53) Modified [Pg. 19, l. 578].**
"... at some sites ..."

*C 54) ll. 497-498: Either use just "b" or "The b parameter".*
**AR 54) Modified [Pg. 19, l. 579].**
"... The b parameter ..."

*C 55) l. 499: Add "the" in front of "temperature factor".*
**AR 55) Added [Pg. 19, l. 580].**
"... the temperature factor ..."

*C 56) l. 501: With "informative" do you mean "sensitive"?*
**AR 56) Indeed. We meant that parameter is sensitive and thus is informative [Pg. 19, l. 583].**

*C 57) l. 502: It should be "favourable conditions".*
**AR 57) Modified [Pg. 19, l. 585].**

*C 58) ll. 504-505: "little impact of environmental stresses due to temperature on GPP during the growing season" -¿ What about outside the growing season? Are the temperature stress parameters just less significant than your phenology submodel parameters? Also, could this not be site-dependent? At some sites, heat might impact GPP during the growing season.*
**AR 58) Indeed what limits the $co_2$ assimilation and gross primary productivity outside of the growing season is the temperature stress. we have revised the text as follows [Pg. 19, l. 583]:**
"In other words, temperature stress limits the $co_2$ assimilation and gross primary productivity outside of the growing season. Phenology parameters play their roles during the growing season. This period indicates favourable condition for plant growth when the temperature stress is mostly not active. Therefore, temperature stress parameters do not significantly influence the modelled GPP. In agreement with our results, Yuan et al. (2007) also reported little impact of environmental stresses due to temperature on GPP during the growing season. It is worth mentioning that the temperature stress is still applied during the growing season, but as the upper-most limits of temperature ($T_{low}$=-2 °C and $T_{high}$=38 °C) do not occur frequently, unless during cold, heat stresses (such as heat years in 2018 and 2019 at the DE-HoH site), the sensitivity of GPP to temperature parameters are less pronounced during the growing season."

*C 59) ll. 506-507: Rephrase to make your point clearer! Unclear what you mean with "a group of daily LAI". Close the bracket after Figure 5.*

**AR 59) As it can be seen from the PCM simulations and in agreement with previous studies, GPP output saturates and becomes insensitive to LAI values above 4 $m^2m^{-2}$. For instance, the simulated LAI at DE-HoH site during the summer period, with maximum LAI usually above 4, an ensemble of LAI's from 4 to 5 $m^2m^{-2}$ correspond to a much narrower resulting GPP at each time step. We modified the sentence to make it clearer [Pg. 19, l. 592].**

"This effect can also be seen in the LAI simulation (e.g., at DE-HoH site) where an ensemble of simulated LAI at each time step during the maturity phase, (i.e., in Figure 7), did not cause much difference in the corresponding GPP output (i.e., in Figure 5)."

*C 60) l. 522: Why "might"? Do they?*

**AR 60) Thanks for the question. In the PCM, the LAI and GPP are simultaneously simulated in the model. Since the SLA is one of the parameters directly related to LAI, while LAI is, in turn, related to resulting GPP, we interpret that when SLA impacts GPP, it can only be through the LAI. However, we use "might" for a further caution. We changed the "might" word to "likely" [Pg. 20, l. 610].**

"Since the LAI output in the model depends on GPP, the studies mentioned above reporting the $SLA$ impact on GPP likely apply for LAI output as well (Li et al., 2016; Arsenault et al., 2018)."

*C 61) l. 527: Add "the" in front of "Fluxnet2015".*
**AR 61) Added [Pg. 20, l. 615].**

*C 62) l. 532: Explain what you mean with "allowing the canopy to reach to its maximum". Instead of "Next important contribution of parameters to the LAI output are those", maybe say something like "Other parameters the LAI output is sensitive to are ..." or "The LAI output is also sensitive to the parameters ..."*

**AR 62) The $L_b$ parameter is the maximum LAI that the ecosystem can sustain. Within PCM the carbon allocation to the leaf pool is maintained until the canopy LAI reaches that maximum value. More detail can be find in (Pasquato et al., 2015). We also modified the sentence [Pg. 20, l. 619]**

"The $L_b$ parameter (Eq. 24), also exhibits a marked sensitivity for the LAI output (Figure

3b) because it directly affects how long carbon allocation to the leaf pool continues until the canopy LAI reach to its maximum value at canopy closure (see (Eq. 26). Other parameters the LAI output is sensitive to are those governing the leaf phenology in the phenology submodel, $L_g$ (Eq. 18), $F_s$ (Eq. 22), $b$ (Eq. 20), $r$ (Eq. 20) (i.e., in Figure 3b)."

*C 63) l. 536: "Lg parameter" -¿ Add "the" or remove "parameter".*
**AR 63) Added [Pg. 20, l. 625].**

*C 64) l. 538: "cold accumulation in degree day" -¿ Below you call it "cold degree days". Choose one name and define what it is! Just say "leaf fall" instead of "the leaf fall event".*
**AR 64) Thank you for the comment. We acknowledge that the sentence was not easy to understand. In fact, the cold accumulation in degree day and cold degree days here do not refer to the name of the parameter but are statements to help describing. To avoid confusion we have removed them from the revised manuscript. $F_s$ (or leaf fall threshold) is a coldness threshold in degree day, below which leaf shed starts (more detail can be found in Yue and Unger (2015)). The phenology submodel in the PCM is adapted from Yue and Unger (2015) who describe it very well. So instead of explaining in more detail, we point the readers to Yue and Unger (2015)). We revised the sentences to avoid confusion [Pg. 20, l. 627].**
"This parameter represents a coldness threshold for leaf fall in degree day. If the cumulative cold degree days from summer solstice (FDD) becomes equal or less than this threshold, then leaves start falling (more detail can be found in (Yue and Unger, 2015)). For instance, a higher threshold would lead to an early leaf shedding, especially in the cold climates where cumulative cold degree days can be reached faster. Therefore, the between site variation of this parameter is not surprising, given the differences in temperature and accumulated cold degree days among study sites. "

*C 65) l. 539: "lower cold degree days accumulation" -¿ "Accumulated cold degree days"? Why does a lower value trigger earlier leaf fall?*
**AR 65) We thank reviewer for this comment. Given the cold degree days accumulation and the negative sign for these values, the word "lower" is changed to "higher". Therefore, A higher value indicates that the accumulated cold degree days (FDD) can reach to or become less than this threshold earlier and leaves start falling. We modified the text [Pg. 21, l. 629.**
"For instance, a higher threshold would lead to an early leaf shedding, especially in the cold climates where cumulative cold degree days can be reached earlier."

*C 66) ll. 546-547: Make clearer what you mean! Why would LAI always decrease, when you change these parameters? Don't you vary the value by +/- 20%?? Also, you say that GPP is less sensitive to these parameters than LAI, but then you explain the sensitivity of*

*LAI by a reduced GPP?*

**AR 66) We agree that the justification looks confusing! We indeed vary the values by ±20%; the LAI can either decrease or increase to reflect these variations. we have added another sentence to the previous one to better clarify this aspect [Pg. 21, l. 638].**

"It might partly be due to the reduced/raised assimilated carbon (GPP) by canopy respiration which, in turn, might decrease/increases the available carbon to be allocated to leaf biomass and affect the resultant LAI. In addition to that, to best of our knowledge, it is the first time that these parameters are thoroughly analysed within a sensitivity analysis framework, and we yet might not be able to find a reason or explanation for this pattern in this study. This calls for future studies to further investigate this aspect."

*C 67) ll. 547-548: How is the sentence "Furthermore, the evaluation of Sobol' indices convergence (see Figure 4) showed relative stability of sensitivity indices at around 8 000 model evaluations." connected to the previous sentences?*

**AR 67) Indeed, it was not the best place for this sentence. It is an independent piece of information about SA. We have therefore replaced this sentence to the end of section 3.1 [Pg. 18, l. 532].**

*C 68) l. 551: Instead of "informative parameters" maybe say "The X most sensitive parameters?"*

**AR 68) We would like to mention that the SA has been conducted for each site individually and the X number of most sensitive parameters at each site vary between 8 to 14. We therefore opted to say informative parameters without attaching specific number to to that retains the generality of the mentioned text. Nevertheless, we have added more clarification in the revised text [Pg. 21, l. 646].**

"... only the most sensitive parameters (depending on the SA result at each site, number of the most sensitive parameters vary between 8 to 14 parameters)..."

*C 69) l. 571: Instead of "validity" maybe use "performance".*

**AR 69) Modified [Pg. 22, l. 667].**

"Taken together, our model exhibits a reasonable performance..."

*C 70) l. 580: "where the model overestimated GPP" -¿ You mention poor performance due to a lack of soil moisture data and a lack of moisture. Why would GPP be overestimated then?*

**AR 70) Thanks for the question. We believe this is due to omitting information on soil water stress. Where soil moisture data is not available (everywhere except the DE-HoH site), soil moisture stress is not accounted for, leading to a overestimation of GPP during times of water limitation. At the DE-Hai site in the late summer 2018, where the model does not account for soil moisture stress, due to unavailability of**

relevant soil moisture information, the GPP is overestimated. The figure below shows the overestimation of GPP at DE-Hai during summer 2018 in a red ellipse.

[Figure]

However, since the soil moisture information was available at the DE-HoH site and the stress factor was applied in the GPP estimation, we were able to compare simulations with and without water stress. When accounting for water stress, the median of GPP output ensembles showed a good agreement with the observed GPP. We also showed that GPP at the DE-HoH site is overestimated without accounting for soil moisture stress factor in the supplemental figure S1.

*C 71) l. 599: Why was the decision made not to include non-structural carbohydrates in the model, if it is specifically made for deciduous broadleaf trees?*
**AR 71) In fact, at the beginning of model set-up, we were not aware of it. Only after analysing the result and looking for reasons for the disagreement, we learnt about non-structural carbohydrates and believe they are the cause of the slightly lagging phase in simulating LAI at the beginning of growing season as compared to the field measurements. Also, note that PCM currently comes without a complete carbon allocation scheme to all pools, which would be a pre-requisite to account for carbon storage. Therefore, in this first version of the PCM the non-structural carbohydrates are not represented in the model. We have acknowledge this part in the manuscript and this leaves the room for further model development.**

*C 72) l. 602: "Eventually" is unnecessary.*
**AR 72:) Modified [Pg. 23, l. 705].**

*C 73) l. 615: "also even" -¿ Use "also" or "even", not both.*
**AR 73) Modified [Pg. 23, l. 718].**

*C 74) Figure 1 caption: It should be "PAR: photosynthetically active radiation". Why do you define certain abbreviations that are in the rectangles but not all of them?*
**AR 74) Thanks. We have modified the "PAR: photosynthetically active radiation" as suggested. All the rectangles are the processes defined in the model except the LUE parameter and Photoperiod variable. Since LUE is one of the most important parameters in the model, we show this parameter in the figure. Also photoperiod is an**

inherent variable and part of autumn phenology which its representation in the figure helps to distinguish between spring and autumn phonology. However, we understand that showing them in rectangles was confusing. We now show these two important variables in ellipse.

"Variables in ellipse show LUE and photoperiod."

*C 75) Table 1: You talked about excluding certain years with missing data. Are these the time periods you actually used?*

**AR 75) Thanks for this reminder! Actually, the time periods in the table show the original time series downloaded from the Fluxnet2015 site. The actual simulation time periods, with exclusion of the first year – first year is needed for the calculation of budburst day of subsequent year –, are the time series shown in the figure S4, S5, and S6 for different sites. We added another column with a name as "Simulation period".**

"US-MMS: 1999-2014, US-Oho: 2004-2013, IT-Ro1: 2001-2006, US-Ha1: 2003-2012, FR-Fon: 2005-2014, DE-Hai: 2000-2018, DE-HoH: 2014-2018, CA-Oas: 1996-2010, DK-Sor: 2006-2013. Please see the response to the AR 9 response to the Referee#2 for more detail."

*C 76) Figure 5 caption: I think you mean "shaded areas". Why do you only have the shading for short periods at certain sites?*

**AR 76) Modified to "shaded area". And thanks for the question. As explained in Section 2.2.4, to account for the predictive uncertainty, we selected an ensemble of model runs (outputs) at each site that lie within the top 5% of all the performance metrics. Based on that, at some sites there may be more and at others less ensemble members. Regarding certain and short period, as it can be obviously seen at the DE-HoH site, it shows more uncertainty during drought periods on 2018 and 2019 and emphasis the role of soil moisture stress factor and associated parameters (namely root distribution coefficient and permanent wilting point parameters). Even small variation in the above mentioned parameters makes a larger difference in resultant GPP.**

*C 77) What ensemble are you talking about? Is it an ensemble of the model output using different parameter values?*

**AR 77) Yes, indeed! Out of a total of 10000 parameter sets sampled from their a priori defined ranges (Table 3) at each site we choose the informative paprmeter sets. Here, the grey shaded area corresponds to the resultant ensemble output members.**

**Referee # 2**

Dear Referee,

Thank you very much for your time and attentions on this work. The comments and suggestions are very useful to improve our manuscript. We paid detailed attention to all comments and have addressed all of them below accordingly. We also would like to thank you for the introducing new papers, they are indeed very interesting and helpful. Please find our response in the supplement.

**General comments**

*Bahrami and colleagues presented a manuscript describing the Parsimonious Canopy Model (PCM v1.0), that estimates gross primary productivity and leaf area index. The manuscript is well written (with some technical notes below) and, in my opinion, useful since the authors provides the code for the PCM in R. I have two main concerns: Please consider including in the title: "Developing a Parsimonious Canopy Model (PCM v1.0) to Predict Forest Gross Primary Productivity and Leaf Area Index on deciduous broad-leaved forest" or something that limits to the actual coverage of the study. Right now, the model only has been tested in this type of ecosystems (with good performance), and the actual title kind of oversells the coverage.*

**We appreciate the reviewer's overall positive assessment to our work. We understand the reviewer remark regarding title and therefore have revised the title that now more clearly state the "deciduous broad-leaved forest" for which we have developed and tested our model. The revised title is:**

"Developing a Parsimonious Canopy Model (PCM v1.0) to predict forest gross primary productivity and leaf area index on deciduous broad-leaved forest"

*The phenology module. It is not clear if the phenology module estimates the start and end of the growing seasons using the warm-up period and then these values are used in the subsequent years. If so, this is a limitation of the model, since the SOS and EOS can be different over the years, influencing the carbon uptake period. At the end the annual sum might be correct/similar, however for incorrect reasons. This should be clearly stated in the limitation of the model, if it is the case.*

**Thanks for these remarks! In fact, the phenology module is run for each year. It uses the number of chill days (it counts days with daily mean temperature less than 5 degree centigrade) from winter solstice of the previous year as a variable which influences the budburst occurrence of each next year. We used the warm up period term referring to the last 10 to 11 days of each previous year that are eventually required for estimating variables in the phenology module for its uninterrupted run in**

the subsequent year. Indeed, what we observed using phenology module was different SOS's and EOS's during the study period at each site. And the start and end of carbon uptake is exactly in accordance with the SOS and EOS in each individual year.

**Specific comments and technical corrections**

*C 1) Please check the use of the expression "e.g.", in the text it is used as "e.g.," with the comma, while in the abstract is not.*
**AR 1) Thanks. We changed "e.g." to "e.g.," over the text.**

*C 2) Please check over the text the use of "$R^2$" in uppercase, it should be in lowercase since it is a 1:1 comparison.*
**AR 2) We modified "$R^2$" to "$r^2$" in the text.**

*C 3) Over the text, please use italics when referring to a parameter (i.e., coefficients/parameters from Table 3).*
**AR 3) Thanks for the comment. We used italics when referring to parameters in the text accordingly.**

*C 4) Epsilon is in Eq. 3, not Eq. 4, please correct.*
**AR 4) Thanks for noticing. We modified the equation number to Eq. 3 [Pg. 7, l. 195].**

*C 5) L217-218. Please check the references in this sentence.*
**AR 5) Thanks. We Modified the sentence [Pg. 9, l. 268].**
"According to Granier et al. (1999) and Fischer et al. (2016) the $scw$ ..."

*C 6) L260. If $f_{SP} = f_{ST}$ (Eq 21), why not make it simple since Eq 18?*
**AR 6) Thanks for your kind suggestion. By keeping this equation, we wanted to be consistent with the autumn phenology (Eq. 25), which is the next part where photoperiod factor $f_{dl}$ also plays a role.**

*C 7) L288. It should be BIOME-BGC (check this all over the text), please check if this version also includes the MUSSO*
**AR 7) Thanks for the comment. We added the latest version as Biome-BGCMuSo v6.2. [Pg. 12, l. 340].**

*C 8) L346-347. How is the PAR-PPFD conversion done?*
**AR 8) Thanks for the question. We use the Fluxnet2015 PPFD variable in $micromol\ m^{-2}\ s^{-1}$ and convert it to PAR in $MJ\ m^{-2}\ day^{-1}$ as following:**

**4.5** $micromol\ m^{-2}\ s^{-1}$ **= 0.000001** $MJ\ m^{-2}\ day^{-1}$**, then multiplied to 86400 to get the corresponding daily values.**
**PAR =PPFD-IN * 0.000001 / 4.5 * 86400**

*C 9) L348-349. Does this mean that the phenology submodel parameters are fixed according to the warm-up year to the subsequent years? Are there implications for using this? Could the authors report the values of the start and end of the growing season for each year of simulation?*

**AR 9) We actually meant that since the phenology module for each individual year needs the number of chilling days from the previous year, the very first year of the observation period cannot be included in the simulations. Instead it is used to determine the budburst day of the first modelling year. For instance if the study period is from 2006 to 2013 then the simulation starts from 2007 to 2013. We acknowledge this sentence was not clear enough. We added to the sentence to make this clearer. In the following we also report the range of simulated Phenological transition dates including the start and end of the growing season (Julian date) for each year of simulation at the DE-HoH site.**

| Year | Start of growing Season | Maturity state | Start of leaf fall | End of growing Season |
|------|------------------------|----------------|--------------------|-----------------------|
| 2015 | 109-123 | 162-186 | 257-270 | 296-298 |
| 2016 | 111-122 | 163-183 | 270 | 297-298 |
| 2017 | 101-129 | 168-183 | 251-270 | 290-298 |
| 2018 | 103-112 | 146-168 | 270-272 | 296-298 |
| 2019 | 105-114 | 154-179 | 271-273 | 297-298 |

"In other words, since the phenology module for each individual year needs the number of chilling days from the previous year, the very first year of observations is not included in the simulations. It is only used for to calculate budburst day of the first simulation year."

*C 10) L476. Please check the references in this sentence.*
**AR 10) Modified [Pg. 18, l. 553].**
"... in similar studies (e.g., Xiao et al. (2004) and Xin et al. (2019)); ..."

*C 11) L486. Please check this reference (Hirmas et al., 2018, Nature) for increasing the discussion on how soil parameters should not be fixed. I liked this! Hirmas, D.R., Giménez, D., Nemes, A. et al. Climate-induced changes in continental-scale soil macroporosity may intensify water cycle. Nature 561, 100–103 (2018). https://doi.org/10.1038/s41586-018-0463-x*
**AR 11) Thanks for providing this reference. We added it to the discussion part of the**

**revised manuscript [Pg. 19, l. 568].**

"Hirmas et al. (2018) also showed that soil retention properties can change in time. For example, climate change may induce rapid changes in the soil macroporosity and the associated soil hydraulic properties. Those may alter the feedback between climate and land surface."

*C 12) L503-504. "Therefore, corresponding parameters do not significantly influence the modelled GPP". This sentence is ambiguous, since I cannot interpret to which parameters the authors are referring to (i.e., temperature stress or phenology).*

**AR 12) Thanks for the comment. What we tried to explain is that in general upon arrival of favourable condition for plant growth, the period between SOS and EOS, temperature stress and the corresponding parameter roles on the GPP is less pronounced. To make this clearer, we have revised the texts [Pg. 19, l. 586].**

"Therefore, temperature stress parameters do not significantly influence the modelled GPP."

*C 13) L508. Please check how the references are used.*

**AR 13) Thanks. We modified the sentence [Pg. 20, l. 595].**

"This is in agreement with the previous studies of Jung et al. (2007) and Lee et al. (2019), which showed that GPP output saturates and becomes insensitive at LAI values above $4\,m^2\,m^{-2}$."

*C 14) L571-583. This might be a good reference (Vargas et al) for the discussion of drought and Mediterranean ecosystems. Vargas, R., Sonnentag, O., Abramowitz, G. et al. Drought Influences the Accuracy of Simulated Ecosystem Fluxes: A Model-Data Meta-analysis for Mediterranean Oak Woodlands. Ecosystems 16, 749–764 (2013). https://doi.org/10.1007/s10021-013-9648-1*

**AR 14) Thanks for this reference. We have added it to the discussion part [Pg. 22, l. 574].**

"Vargas et al. (2013), also discussed inter-annual dynamics of soil moisture effect on GPP flux in Mediterranean ecosystems using five process-oriented ecosystem models including water balance. They observed a systematically underestimation of GPP in the models that were accounting for soil water balance. Those underestimations may have been related to the complex nature of Mediterranean ecosystems, e.g., due to deep roots and an important role of the lower canopy. In contrast, here we overestimate the GPP and believe that this is due to lack of local information on soil moisture stress. More information of soil moisture stress is therefore expected to improve the model. Overall, they emphasize the importance of drought conditions and the complex nature of Mediterranean ecosystems in representing forest dynamics, including GPP flux."

*C 15) In Tables 4-5, I recommend to the authors to report the linear regression coefficients*

*(slope and intercept), not only RMSE and r², so the reader can know the biases.*
**AR 15) The linear regression coefficients are now added to the tables.**

[revised manuscript text omitted]

---

## Author Comment (AC2)

Dear Editor,

We wish to thank you and the referees for your precious time in reviewing our paper and providing valuable comments. It was your valuable and insightful comments that led to possible improvements in the current version. Following the editorial office suggestions, we combined the Supplementary figure's captions and the figures themselves into a single pdf supplement file, we also checked and revised the figures to ensure that figures are accessible to colorblind readers.
We have carefully considered the comments and tried our best to address every one of them. Below we provide the point-by-point responses to referees' comments. Texts in italic are the referees' comments (C), those in black bold style are our responses (AR), and texts marked in red are relevant changes in the manuscript. A marked-up version, showing the changes in the revised manuscripts, has also been prepared to be submitted accordingly. The page and line numbers in this letter refer to the marked-up version. We hope that you will find the changes satisfactory.

Sincerely,

Bahar Bahrami on behalf of the co-authors
bahreh.bahrami@ufz.de
Department of Computational Hydrosystems
UFZ, Leipzig, Germany

**Referee # 2**

Dear Referee,

Thank you very much for your time and attentions on this work. The comments and suggestions are very useful to improve our manuscript. We paid detailed attention to all comments and have addressed all of them below accordingly. We also would like to thank you for the introducing new papers, they are indeed very interesting and helpful. Please find our response in the supplement.

**General comments**

*Bahrami and colleagues presented a manuscript describing the Parsimonious Canopy Model (PCM v1.0), that estimates gross primary productivity and leaf area index. The manuscript*

*is well written (with some technical notes below) and, in my opinion, useful since the authors provides the code for the PCM in R. I have two main concerns: Please consider including in the title: "Developing a Parsimonious Canopy Model (PCM v1.0) to Predict Forest Gross Primary Productivity and Leaf Area Index on deciduous broad-leaved forest" or something that limits to the actual coverage of the study. Right now, the model only has been tested in this type of ecosystems (with good performance), and the actual title kind of oversells the coverage.*

**We appreciate the reviewer's overall positive assessment to our work. We understand the reviewer remark regarding title and therefore have revised the title that now more clearly state the "deciduous broad-leaved forest" for which we have developed and tested our model. The revised title is:**

"Developing a Parsimonious Canopy Model (PCM v1.0) to predict forest gross primary productivity and leaf area index on deciduous broad-leaved forest"

*The phenology module. It is not clear if the phenology module estimates the start and end of the growing seasons using the warm-up period and then these values are used in the subsequent years. If so, this is a limitation of the model, since the SOS and EOS can be different over the years, influencing the carbon uptake period. At the end the annual sum might be correct/similar, however for incorrect reasons. This should be clearly stated in the limitation of the model, if it is the case.*

**Thanks for these remarks! In fact, the phenology module is run for each year. It uses the number of chill days (it counts days with daily mean temperature less than 5 degree centigrade) from winter solstice of the previous year as a variable which influences the budburst occurrence of each next year. We used the warm up period term referring to the last 10 to 11 days of each previous year that are eventually required for estimating variables in the phenology module for its uninterrupted run in the subsequent year. Indeed, what we observed using phenology module was different SOS's and EOS's during the study period at each site. And the start and end of carbon uptake is exactly in accordance with the SOS and EOS in each individual year.**

**Specific comments and technical corrections**

*C 1) Please check the use of the expression "e.g.", in the text it is used as "e.g.," with the comma, while in the abstract is not.*
**AR 1) Thanks. We changed "e.g." to "e.g.," over the text.**

*C 2) Please check over the text the use of "$R^2$" in uppercase, it should be in lowercase since it is a 1:1 comparison.*
**AR 2) We modified "$R^2$" to "$r^2$" in the text.**

*C 3) Over the text, please use italics when referring to a parameter (i.e., coefficients/parameters from Table 3).*

**AR 3) Thanks for the comment. We used italics when referring to parameters in the text accordingly.**

*C 4) Epsilon is in Eq. 3, not Eq. 4, please correct.*

**AR 4) Thanks for noticing. We modified the equation number to Eq. 3 [Pg. 7, l. 195].**

*C 5) L217-218. Please check the references in this sentence.*

**AR 5) Thanks. We Modified the sentence [Pg. 9, l. 268].**

"According to Granier et al. (1999) and Fischer et al. (2016) the $scw$ ..."

*C 6) L260. If $f_{SP}$ = $f_{ST}$ (Eq 21), why not make it simple since Eq 18?*

**AR 6) Thanks for your kind suggestion. By keeping this equation, we wanted to be consistent with the autumn phenology (Eq. 25), which is the next part where photoperiod factor $f_{dl}$ also plays a role.**

*C 7) L288. It should be BIOME-BGC (check this all over the text), please check if this version also includes the MUSSO*

**AR 7) Thanks for the comment. We added the latest version as Biome-BGCMuSo v6.2. [Pg. 12, l. 340].**

*C 8) L346-347. How is the PAR-PPFD conversion done?*

**AR 8) Thanks for the question. We use the Fluxnet2015 PPFD variable in $micromol\ m^{-2}\ s^{-1}$ and convert it to PAR in $MJ\ m^{-2}\ day^{-1}$ as following:**

**4.5 $micromol\ m^{-2}\ s^{-1}$ = 0.000001 $MJ\ m^{-2}\ day^{-1}$, then multiplied to 86400 to get the corresponding daily values.**

**PAR =PPFD-IN * 0.000001 / 4.5 * 86400**

*C 9) L348-349. Does this mean that the phenology submodel parameters are fixed according to the warm-up year to the subsequent years? Are there implications for using this? Could the authors report the values of the start and end of the growing season for each year of simulation?*

**AR 9) We actually meant that since the phenology module for each individual year needs the number of chilling days from the previous year, the very first year of the observation period cannot be included in the simulations. Instead it is used to determine the budburst day of the first modelling year. For instance if the study period is from 2006 to 2013 then the simulation starts from 2007 to 2013. We acknowledge this sentence was not clear enough. We added to the sentence to make this clearer. In the**

**following we also report the range of simulated Phenological transition dates including the start and end of the growing season (Julian date) for each year of simulation at the DE-HoH site.**

| Year | Start of growing Season | Maturity state | Start of leaf fall | End of growing Season |
|------|------------------------|----------------|--------------------|-----------------------|
| 2015 | 109-123 | 162-186 | 257-270 | 296-298 |
| 2016 | 111-122 | 163-183 | 270 | 297-298 |
| 2017 | 101-129 | 168-183 | 251-270 | 290-298 |
| 2018 | 103-112 | 146-168 | 270-272 | 296-298 |
| 2019 | 105-114 | 154-179 | 271-273 | 297-298 |

"In other words, since the phenology module for each individual year needs the number of chilling days from the previous year, the very first year of observations is not included in the simulations. It is only used for to calculate budburst day of the first simulation year."

*C 10) L476. Please check the references in this sentence.*
**AR 10) Modified [Pg. 18, l. 553].**
"... in similar studies (e.g., Xiao et al. (2004) and Xin et al. (2019)); ..."

*C 11) L486. Please check this reference (Hirmas et al., 2018, Nature) for increasing the discussion on how soil parameters should not be fixed. I liked this! Hirmas, D.R., Giménez, D., Nemes, A. et al. Climate-induced changes in continental-scale soil macroporosity may intensify water cycle. Nature 561, 100–103 (2018). https://doi.org/10.1038/s41586-018-0463-x*
**AR 11) Thanks for providing this reference. We added it to the discussion part of the revised manuscript [Pg. 19, l. 568].**
"Hirmas et al. (2018) also showed that soil retention properties can change in time. For example, climate change may induce rapid changes in the soil macroporosity and the associated soil hydraulic properties. Those may alter the feedback between climate and land surface."

*C 12) L503-504. "Therefore, corresponding parameters do not significantly influence the modelled GPP". This sentence is ambiguous, since I cannot interpret to which parameters the authors are referring to (i.e., temperature stress or phenology).*
**AR 12) Thanks for the comment. What we tried to explain is that in general upon arrival of favourable condition for plant growth, the period between SOS and EOS, temperature stress and the corresponding parameter roles on the GPP is less pronounced. To make this clearer, we have revised the texts [Pg. 19, l. 586].**
"Therefore, temperature stress parameters do not significantly influence the modelled GPP."

*C 13) L508. Please check how the references are used.*

**AR 13) Thanks. We modified the sentence [Pg. 20, l. 595].**

"This is in agreement with the previous studies of Jung et al. (2007) and Lee et al. (2019), which showed that GPP output saturates and becomes insensitive at LAI values above $4\ m^2\ m^{-2}$."

*C 14) L571-583. This might be a good reference (Vargas et al) for the discussion of drought and Mediterranean ecosystems. Vargas, R., Sonnentag, O., Abramowitz, G. et al. Drought Influences the Accuracy of Simulated Ecosystem Fluxes: A Model-Data Meta-analysis for Mediterranean Oak Woodlands. Ecosystems 16, 749–764 (2013). https://doi.org/10.1007/s10021-013-9648-1*

**AR 14) Thanks for this reference. We have added it to the discussion part [Pg. 22, l. 574].**

"Vargas et al. (2013), also discussed inter-annual dynamics of soil moisture effect on GPP flux in Mediterranean ecosystems using five process-oriented ecosystem models including water balance. They observed a systematically underestimation of GPP in the models that were accounting for soil water balance. Those underestimations may have been related to the complex nature of Mediterranean ecosystems, e.g., due to deep roots and an important role of the lower canopy. In contrast, here we overestimate the GPP and believe that this is due to lack of local information on soil moisture stress. More information of soil moisture stress is therefore expected to improve the model. Overall, they emphasize the importance of drought conditions and the complex nature of Mediterranean ecosystems in representing forest dynamics, including GPP flux."

*C 15) In Tables 4-5, I recommend to the authors to report the linear regression coefficients (slope and intercept), not only RMSE and $r^2$, so the reader can know the biases.*

**AR 15) The linear regression coefficients are now added to the tables.**

**References**

Fischer, R., Bohn, F., Dantas de Paula, M., Dislich, C., Groeneveld, J., Gutiérrez, A. G., Kazmierczak, M., Knapp, N., Lehmann, S., Paulick, S., Pütz, S., Rödig, E., Taubert, F., Köhler, P., and Huth, A.: Lessons learned from applying a forest gap model to understand ecosystem and carbon dynamics of complex tropical forests, Ecological Modelling, 326, 124–133, https://doi.org/https://doi.org/10.1016/j.ecolmodel.2015.11.018, next generation ecological modelling, concepts, and theory: structural realism, emergence, and predictions, 2016.

Granier, A., Bréda, N., Biron, P., and Villette, S.: A lumped water balance model to evaluate duration and intensity of drought constraints in forest stands, Ecological Modelling, 116, 269–283, https://doi.org/https://doi.org/10.1016/S0304-3800(98)00205-1, 1999.

Hirmas, D., Giménez, D., Nemes, A., Kerry, R., Brunsell, N., and Wilson, C.: Climate-induced changes in continental-scale soil macroporosity may intensify water cycle, Nature, 561, https://doi.org/10.1038/s41586-018-0463-x, 2018.

Jung, M., Vetter, M., Herold, M., Churkina, G., Reichstein, M., Zaehle, S., Ciais, P., Viovy, N., Bondeau, A., Chen, Y., Trusilova, K., Feser, F., and Heimann, M.: Uncertainties of modeling gross primary productivity over Europe: A systematic study on the effects of using different drivers and terrestrial biosphere models, Global Biogeochemical Cycles, 21, https://doi.org/10.1029/2006GB002915, 2007.

Lee, H., Park, J., Cho, S., Lee, M., and Kim, H.: Impact of leaf area index from various sources on estimating gross primary production in temperate forests using the JULES land surface model, Agricultural and Forest Meteorology, p. 107614, 2019.

Vargas, R., Sonnentag, O., Abramowitz, G., Carrara, A., Chen, J., Ciais, P., Correia, A., Keenan, T., Kobayashi, H., Ourcival, J., Papale, D., Pearson, D., Pereira, J., Piao, S., Rambal, S., and Baldocchi, D.: Drought Influences the Accuracy of Simulated Ecosystem Fluxes: A Model-Data Meta-analysis for Mediterranean Oak Woodlands, Ecosystems, 16, 749–764, https://doi.org/10.1007/s10021-013-9648-1, 2013.

Xiao, X., Zhang, Q., Braswell, B., Urbanski, S., Boles, S., Wofsy, S., Moore, B., and Ojima, D.: Modeling gross primary production of temperate deciduous broadleaf forest using satellite images and climate data, Remote Sensing of Environment, 91, 256–270, https://doi.org/https://doi.org/10.1016/j.rse.2004.03.010, 2004.

Xin, Q., Dai, Y., and Liu, X.: A simple time-stepping scheme to simulate leaf area index, phenology, and gross primary production across deciduous broadleaf forests in the eastern United States, Biogeosciences, 16, 467–484, https://doi.org/10.5194/bg-16-467-2019, 2019.